# "*very likely*" Means "*uncertain*"? How LLMs Diverge from Humans in Linguistic Uncertainty Quantification

Jinhao Duan [* 1]   Zicheng Liu [* 2]   Zijie Liu [* 1]   Kaidi Xu [3]   Tianlong Chen [1]

## Abstract

Humans express uncertainty verbally via markers (e.g., "possible," "likely"), yet most LLM uncertainty quantification (UQ) relies on costing likelihood- or consistency-based signals. From a cognitive perspective, accurate verbal uncertainty reflects metacognitive monitoring, representing knowledge boundaries ("knowing that you don't know") to support regulation and information seeking. In this paper, we investigate *How LLMs diverge from humans in verbal uncertainty quantification? Can verbal markers reliably quantify LLM uncertainty?* We curate a corpus of human uncertainty markers from psychology and decision-science literature and benchmark LLMs against it. We observe that LLMs encode verbal uncertainty with numerical levels that differ substantially from those of humans. We then introduce VOCAL, a novel optimization-based algorithm that learns an optimal uncertainty profile over uncertainty markers directly from LLM outputs. By fitting a marker–uncertainty mapping to best explain empirical correctness, VOCAL discovers how much probability mass each verbal marker should convey, rather than estimating uncertainty via repeated sampling. VOCAL enables a direct, marker-level comparison of confidence semantics between humans and LLMs, disentangling mismatch and revealing systematic confidence disparities in verbal expressions.

## 1. Introduction

Despite large language models'(LLMs) recent remarkable success across diverse domains (Xie et al., 2023; Colombo et al., 2024; Yang et al., 2024; Thapa et al., 2025), a fundamental question remains: when should we trust LLMs' responses? This question highlights the need to make LLMs more trustworthy and responsible. Hallucinations are not only mistakes but also risks that can reduce users' trust and cause harm in sensitive applications (Asgari et al., 2025; Das et al., 2025), like giving unsafe treatment advice in biomedicine. One promising approach to mitigating this phenomenon is uncertainty quantification (UQ) (Malinin & Gales, 2020; Kuhn et al., 2023a; Duan et al., 2024), which provides a probabilistic signal to estimate trustworthiness without labeled data. This capability facilitates distinguishing valid predictions from hallucinations or extrapolation, essential for real-world deployment.

However, existing approaches for quantifying hallucination in LLMs still have some limitations, primarily dividing into two main groups: sampling-based techniques (Farquhar et al., 2024; Kossen et al., 2024; Li et al., 2025a; McCabe et al., 2025) and logits-based techniques (Sriramanan et al., 2024; Nguyen et al., 2025; Ma et al., 2025; Yang et al., 2025). While logits-based methods such as Predictive Entropy (PE) (Malinin & Gales, 2020) offer reliability, they suffer from lexical sensitivity and computational overhead. Sampling-based methods, such as semantic entropy (SE) (Kuhn et al., 2023b), ensemble variance, or consistency checks across multiple generations, can be more robust but are also slow and costly. Therefore, a recent direction focuses on verbal uncertainty, where models output numerical uncertainty scores (Tian et al., 2023). Although improving calibration compared to raw probability outputs, precise quantification remains unnatural in contrast to qualitative terms such as "*possible*," "*likely*," or "*almost certain*" that better capture the nuance of human reasoning. This contrast shows a gap in current methods and points to the need for approaches that allow models to express uncertainty in a way that is more natural, human-like, and trustworthy for real-world use ( Figure 1 (left)).

Motivated by this gap, we ask:

> *How LLMs diverge from humans in verbal uncertainty quantification? Can verbal markers reliably quantify LLM uncertainty?*

[1]Department of Computer Science, UNC-Chapel Hill [2]Department of Mathematics, The University of Hong Kong [3]Department of Data Science, City University of Hong Kong. Correspondence to: Tianlong Chen <tianlong@cs.unc.edu>.

*Proceedings of the $43^{rd}$ International Conference on Machine Learning*, Seoul, South Korea. PMLR 306, 2026. Copyright 2026 by the author(s).

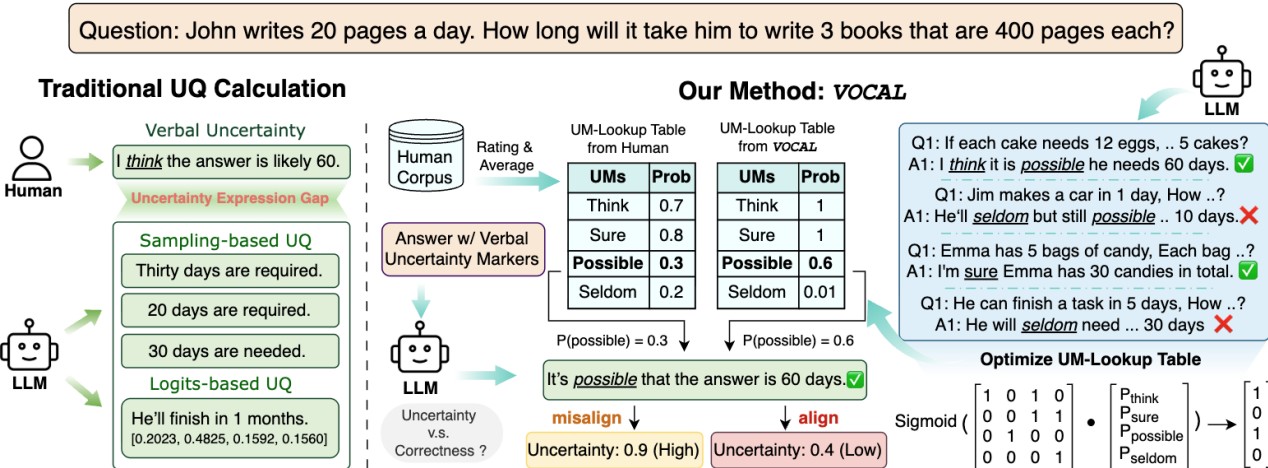

*Figure 1.* Comparison of traditional uncertainty quantification (UQ) methods and our method VOCAL. Traditional UQ methods (sampling-based and logits-based) exhibit a gap with human uncertainty expressions. In VOCAL, UM-lookup tables derived from human data alone cannot fully capture model uncertainty, so they are optimized with the model's confidence distribution to better align with its internal uncertainty expressions.

To study this question, we first construct the verbal uncertainty marker lookup table (UM-Lookup) that maps qualitative expressions of uncertainty to numerical representations. The lookup table is built through a literature review grounded in psychology and decision science (Lichtenstein & Newman, 1967; Beyth-Marom, 1982; Wesson & Pulford, 2009), followed by a debiasing procedure to refine ambiguous cases. We then aggregate judgments from more than 300 human annotators, resulting in a curated resource of 115 distinct verbal uncertainty markers with associated numeric interpretations. Leveraging this resource, we evaluate the ability of LLMs to align their verbal expressions of uncertainty with human interpretations. Our results show that LLMs demonstrate non-trivial UQ performance when assessed against the UM-Lookup. For example, when evaluated with GPT-4o (Achiam et al., 2023) model on SciQ dataset (Welbl et al., 2017), verbal uncertainty outperforms representative logits-based and sampling-based methods such as PE and SE, achieving an improvement of 4.7% AUROC and 5.6% AUROC, respectively. However, across broader benchmarks, verbalized UQ remains weaker than strong UQ baselines, reflecting a gap between human-derived lookup tables for uncertainty markers and LLM uncertainty signals.

This gap largely arises from divergent interpretations: LLMs often associate terms like *"possible"* with significantly lower uncertainty than humans. Furthermore, unlike humans often combine multiple verbal uncertainty markers to convey more fine-grained or complex levels of uncertainty, LLMs typically rely on single markers at each time (Vogel et al., 2022). These differences suggest a gap between human communication patterns and how LLMs currently express verbal uncertainty. To address this gap, we propose VOCAL,

an approximation algorithm that provides an optimal mapping solution between uncertainty markers and numerical uncertainty levels by adapting to the uncertainty distribution of each model ( Figure 1 (right)). VOCAL is evaluated over comprehensive experiments on a wide range of models and datasets. Our results demonstrate that the VOCAL significantly outperforms single-sample UQ methods, such as (Aichberger et al., 2025), and achieve comparable performance as multi-sample UQs, without additional sampling or computational requests. Our contributions are:

- We present a lookup table that maps human verbal uncertainty markers to numerical uncertainty scores, grounded in psychology and decision science. This lookup table is a foundational resource that could not only benefit follow-up verbal uncertainty quantification methods but also inspire the metacognition monitoring research in the future.

- We propose a simple yet effective method, VOCAL, that optimizes the alignment between verbal markers and model uncertainty distributions.

- We conduct comprehensive experiments across multiple models and datasets, providing in-depth analysis and demonstrating the effectiveness of our method. We demonstrate that VOCAL significantly outperforms single-sample UQ methods and achieves comparable performances as multi-sample UQ methods, with significantly reduced computational cost.

## 2. Related Work

**Sampling-Based LLM Uncertainty Quantification.** The need to mitigate untrustworthy outputs from LLMs, such

as hallucinations, has made UQ a critical area of research. UQ for free-form generative models is uniquely challenging because a correct answer can be expressed in countless semantically equivalent ways (Lin et al., 2023; Kuhn et al., 2023a). This renders early methods like predictive entropy (PE) insufficient, as they often misinterpret this benign lexical variance as genuine semantic uncertainty (Kuhn et al., 2023a). To address this, a significant body of work has shifted towards semantic-aware UQ. Semantic Entropy (SE) (Kuhn et al., 2023a) clusters equivalent outputs to estimate uncertainty, while Semantic Density (SD) (Qiu & Miikkulainen, 2024) quantifies a response's uncertainty by measuring its density within a semantic space. In contrast, other methods probe the internal states or consistency of the LLM. Deg (Lin et al., 2023) and its successor INSIDE (Chen et al., 2024) analyze consistency across multiple generations to quantify uncertainty from a black-box perspective. Furthermore, Shifting Attention to Relevance (SAR) (Duan et al., 2024) addresses the generative imbalance by assigning more weight to semantically relevant parts of a generation. In more complex scenarios, UProp (Duan et al., 2025) introduces a framework to decompose and quantify uncertainty propagation in multi-step decision processes. Alternatively, G-NLL (Aichberger et al., 2025) offers a computationally efficient UQ method based on the negative log-likelihood of a single greedy-decoded output, challenging the necessity of multi-sampling. These diverse approaches highlight the evolution of LLM UQ from simple lexical metrics to more semantically robust, context-aware, and computationally efficient solutions.

**Linguistic Uncertainty Quantification.** Verbalized uncertainty, which employs natural language to articulate uncertainty, was pioneered by Mielke et al. (2022); Lin et al. (2022). Early black-box evaluations demonstrated that inherent overconfidence can be mitigated via carefully designed prompts and aggregation methods (Tian et al., 2023; Xiong et al., 2023). To achieve anthropomimetic uncertainty, in which models emulate nuanced human expression to enhance trust (Ulmer et al., 2025), recent approaches actively optimize alignment via supervised fine-tuning and RLHF (Chaudhry et al., 2024; Liu et al., 2024; Leng et al., 2025; Li et al., 2025b). Despite these optimization efforts, empirical studies reveal a persistent faithfulness gap. Ji et al. (2025) and Yona et al. (2024) identify that verbal decisiveness often diverges from intrinsic probabilities or internal verbal uncertainty features (VUF), leading to confident hallucinations. However, Band et al. (2024); Yoon et al. (2025) observe that utilizing chain-of-thought reasoning can improve calibration trajectories during long-form generation. Zhou et al. (2023a) attribute sensitivity to pretraining mimicry, where markers often signal ignorance, while Belem et al. (2024) find numerical interpretations biased by encoded priors. Extending this to human-AI in-

teraction, Kim et al. (2024b) demonstrates that the specific phrasing of these uncertainty expressions modulates user reliance and trust in decision-making tasks.

Instead of expensive fine-tuning, we treat verbal uncertainty as an inherent predictive signal. We propose VOCAL, which optimizes uncertainty mappings via a lightweight lookup table without parameter updates. Unlike sampling-based methods, VOCAL extracts optimized uncertainty from a single response, offering a computationally efficient UQ alternative.

# 3. Preliminary: How LLMs Diverge from Humans in Verbal Uncertainty

## 3.1. Problem Statement: Uncertainty Quantification

Uncertainty quantification (UQ) aims to measure the degree of doubt that a model exhibits with respect to its generations. In the context of LLMs, UQ evaluates the doubt that an LLM parameterized by $\theta$ assigns to a generation $y \sim p_\theta(y \mid x)$, given an input $x$. Formally, let $\mathcal{Q}$ denote a UQ method. The corresponding uncertainty score $q$ associated with $y$ is defined as $q = \mathcal{Q}(y, x, \theta) \in \mathbb{R}$. The specific realization of $\mathcal{Q}$ varies across different UQ approaches, depending on the underlying assumptions and techniques employed. In Section A, we present the realizations of popular LLM UQ methods in detail.

**Performance Evaluation.** The performance evaluation of UQ usually follows a "correctness prediction" manner, measuring the correlation between the calculated uncertainty score from a UQ method $\mathcal{Q}$ and the correctness of model generations, with metrics such as AUROC and hallucination detection accuracy. A higher AUROC or detection accuracy means $\mathcal{Q}$ correctly predicts the correctness of model generations, indicating a good uncertainty estimator.

## 3.2. Human Verbal Uncertainty and its Numerical Representation

Humans usually express their uncertainty in verbal form, with uncertainty markers (UMs) such as "*might*", or "*probably*", which encode a speaker's degree of uncertainty. Formally, we denote by $\mathcal{Q}_{VU}$ a UQ that quantifies uncertainty from UMs. Then, given a model generation $y$, its verbal uncertainty $q$ is denoted by $q_y = \mathcal{Q}_{VU}(\mathcal{V}_y)$, where $\mathcal{U}_y = \{u_1, u_2, \cdots\}$ are the extracted UMs from $y$. However, there are two challenges blocking the quantitative evaluation: ① *How to convert human UMs to numerical representations?* , even though we obtained their numerical scores, ② *how to aggregate numerical scores from multiple UMs?*

To address these challenges, we introduce the first large-scale lookup table of human uncertainty, UM-Lookup ta-

ble, that maps human UMs to numerical probabilities. Our `UM-Lookup` is grounded in foundational empirical studies from psychology and decision science, including the seminal works of Lichtenstein & Newman (1967), Beyth-Marom (1982), Wesson & Pulford (2009), and the comprehensive meta-analysis by Vogel et al. (2022). Statistically, we collect 115 unique UMs, with each phrase's value derived from an average of 336 human ratings. This process yields a standardized confidence scale on a probabilistic $[0, 1]$ range, containing expressions like "impossible" (0.0), "tossup" (0.50), and "definite" (0.99). To remove the bias during the aggregation, we standardize the varied data formats from these sources, via direct probability estimates (Lichtenstein & Newman, 1967), numerical ranges (Beyth-Marom, 1982), Likert scales (Wesson & Pulford, 2009), and meta-analytic weighted means (Vogel et al., 2022), resulting in a consistent structure of a phrase, its mean value, and its frequency (N). The detailed methodology for this normalization and aggregation, along with the complete human VUE lookup table, is provided in Section B. With the `UM-Lookup`, each UM could be effectively converted to a numerical representation.

In terms of the aggregation strategy of multiple UMs, empirical work shows that when people use multiple verbal probability terms in one statement, listeners (and coders) tend to average them into a single "middle" probability (Budescu & Wallsten, 1995). Thus, we simply average all the `UM-Lookup`(UMs) as the final quantified uncertainty:

$$q_{\boldsymbol{y}} = \mathcal{Q}_{\text{VU-H}}(\mathcal{V}_{\boldsymbol{y}}) = \frac{1}{N} \sum_i \left(1 - \text{UM-Lookup}(\boldsymbol{u}_i)\right),$$

where $N$ is the number of UMs from $\boldsymbol{y}$ and $\boldsymbol{u}_i$ is the $i$-th UM in $\mathcal{V}_{\boldsymbol{y}}$. We use $(1 - \text{UM-Lookup}(\boldsymbol{u}_i)$ to convert from confidence to uncertainty. In the rest of this paper, we denote by $\mathcal{Q}_{\text{VU-H}}$ the verbal UQ method equipped with human verbal uncertainty mapping `UM-Lookup`.

### 3.3. Analytical Insights

We evaluate GPT-4o (Achiam et al., 2023) and DeepSeek-V3.1 (DeepSeek-AI, 2024) over diverse datasets, such as GSM-Hard (Gao et al., 2022), GSM8K (Cobbe et al., 2021), MedQA (Jin et al., 2020), PIQA (Bisk et al., 2020). We prompt LLMs to express verbal uncertainty and quantify uncertainty via $\mathcal{Q}_{\text{VU-H}}$. Specifically, we use two five-shot strategies: a standard Chain-of-Thought(CoT) prompting (Wei et al., 2023) and CoT with verbal uncertainty prompting, where the latter incorporates the UM list (see Section C for details). In Section F.1, we demonstrate that verbal uncertainty maintains general performance as the CoT. As illustrated in Figure 7, we evaluate model accuracy under both our verbal uncertainty prompting and a standard CoT baseline. Across all evaluated models on the GSM8K dataset, from GPT-4o to Llama-3.2-3B-Instruct, performance remains on par, with no statistically significant

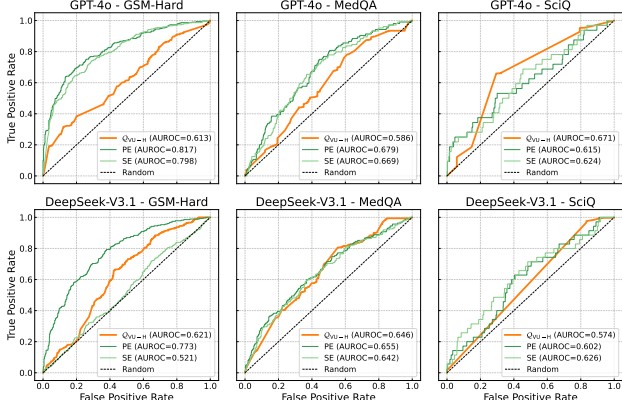

*Figure 2.* The results of verbal uncertainty quantification $\mathcal{Q}_{\text{VU-H}}$ with `UM-Lookup` table collected from human. $\mathcal{Q}_{\text{VU-H}}$ achieves non-trivial UQ performance in many cases, indicating that LLMs share similar uncertainty expression as humans to a certain degree.

degradation in accuracy. This result provides an important validation: the elicitation of verbal uncertainty does not impose a significant performance penalty, thereby preserving the models' core problem-solving efficacy.

$\mathcal{Q}_{\textbf{VU-H}}$ **Achieves Non-Trivial UQ Performance** In Figure 2, our primary finding is that quantifying uncertainty via a human-sourced verbal lookup table, $\mathcal{Q}_{\text{VU-H}}$, provides a meaningful signal for UQ. This method achieves non-trivial performance (where AUROC is significantly greater than 0.5) in 7 out of the 8 evaluated model-dataset configurations. In several cases, its performance is highly competitive with or even surpasses popular UQ baselines. For instance, with GPT-4o on the SciQ dataset, $\mathcal{Q}_{\text{VU-H}}$ outperforms both Probability Entropy (PE) and Semantic Entropy (SE). Similarly, for DeepSeek-V3.1 on MedQA, our method's performance is on par with both baselines.

However, we also identify clear limitations. While often effective, $\mathcal{Q}_{\text{VU-H}}$ is frequently outperformed by PE and can fail notably, such as with GPT-4o on GSM8K, where its AUROC falls below random chance. We attribute these mixed results to a fundamental discrepancy: the uncertainty score assigned to a UM via our human-source `UM-Lookup` table does not always match the LLM's internal uncertainty state at the moment it generates that expression.

**Advanced LLMs Express More Diverse Uncertainty Expressions.** With proper prompting, we find that advanced LLMs can express a diverse and frequent set of verbal uncertainty markers. As shown in Figure 3, large-scale models such as GPT-4o and DeepSeek-V3.1 achieve the highest diversity scores (calculated by entropy). Results are obtained on the GSM8K dataset. Conversely, smaller models demonstrate a limited capacity for expressing nuanced uncertainty. This tendency is consistent with the well-documented challenge of overconfidence in LLMs (Jiang et al., 2021; Xiong

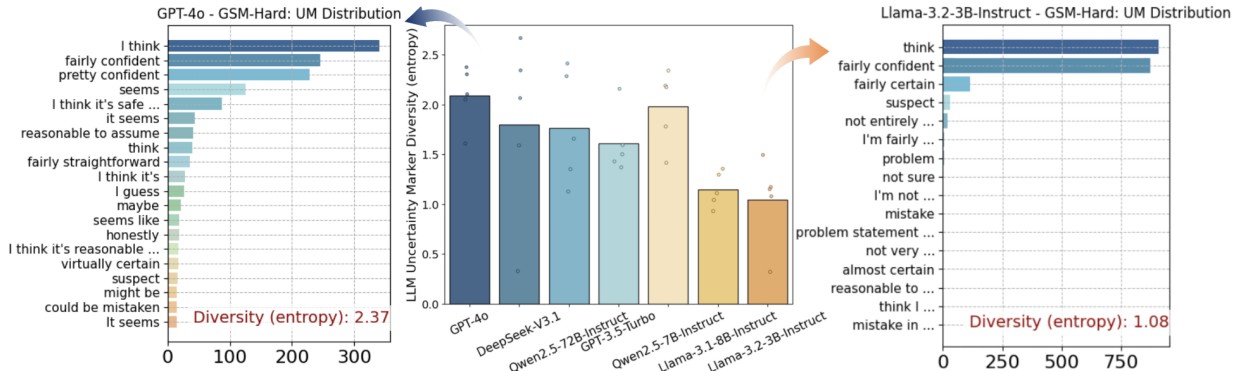

*Figure 3.* The distributions of uncertainty markers expressed by LLMs. We show that advanced LLMs, such as GPT-4o, express uncertainty in a more diverse manner compared to small LLMs (e.g., Llama-3.1-8B-Instruct and Llama-3.2-3B-Instruct). This also reveals that small LLMs tend to be over confident.

et al., 2023; Tian et al., 2023). Such overconfidence is a critical issue, as it can lead to significant errors (Zhou et al., 2023b), reduce user trust (Kim et al., 2024a), and result in harmful downstream consequences (Li, 2023). The complete distributions for all evaluated models are provided in Section F.2.

## 4. VOCAL: Optimizing the Numerical Scores of Verbal Uncertainty Markers for LLMs

In Section 3.3, we observe that although the human-derived verbal uncertainty lookup table (UM-Lookup) provides nontrivial UQ performance, it often lags behind logit- and sampling-based baselines. This naturally raises an important question: rather than relying solely on human estimates, can we instead learn UM-Lookup that are tailored to LLMs themselves?

### 4.1. Setup

To achieve an LLM-tailored probabilistic UM-Lookup, we introduce VOCAL, a simple yet effective algorithm that optimizes the numerical levels of uncertainty markers for LLMs. VOCAL is a data-driven method that learns appropriate uncertainty scores from model generations. To obtain reliable estimations of these scores, we first collect diverse generations across multiple domains, such as mathematics (GSM8K, GSM-Hard), science (PIQA, SciQ), and the medical domain (MedQA). We then apply a verbal uncertainty prompting strategy (see Appendix Section C for detailed templates) to elicit responses with explicit verbal uncertainty expressions and extract UMs together with the correctness of the corresponding generations.

### 4.2. Optimized Uncertainty Levels for LLMs

Formally, we denote by $\mathcal{U} = \{\boldsymbol{u}_1, \boldsymbol{u}_2, \ldots, \boldsymbol{u}_N\}$ the intended UM set extracted from LLM generations. The opti-

mization objective of VOCAL is to learn a suitable numerical score mapping $c_i$ for each UM $u_i$. Formally, given a LLM generation $\boldsymbol{y}$, the aggregated verbal uncertainty of $\boldsymbol{y}$ is then given by $q_{\boldsymbol{y}} = Q_{\text{VU-L}}(V_{\boldsymbol{y}}) = \frac{1}{N_{\boldsymbol{y}}} \sum_{i=1}^{N_{\boldsymbol{y}}}(1 - c_i)$, where $N_{\boldsymbol{y}}$ is the number of UMs in $\boldsymbol{y}$ and $Q_{\text{VU-L}}$ denotes the LLM-specific verbal uncertainty quantifier. $\boldsymbol{u}_{\boldsymbol{y},i} \in \mathcal{U}$ is the $i$-th UM in $\boldsymbol{y}$. The objective of VOCAL is to optimize $c_i$ so that $q_{\boldsymbol{y}}$ faithfully reflects the uncertainty of the LLM with respect to its generation $\boldsymbol{y}$, in particular assigning higher uncertainty to incorrect generations and lower uncertainty to correct generations.

Then, the optimization objective of VOCAL can be formalized in a Binary Cross-Entropy (BCE) manner:

$$\mathcal{L}(\boldsymbol{c}) = \min_{\boldsymbol{c}} \ \mathbb{E}_{(\boldsymbol{x},\boldsymbol{y})}\Big[ -z \log \boldsymbol{c}_{\boldsymbol{y}} - (1-z)\log(1 - \boldsymbol{c}_{\boldsymbol{y}})\Big],$$

where $\mathbf{c}$ denotes the learnable uncertainty assignments for all markers, $\boldsymbol{c}_{\boldsymbol{y}} = \frac{1}{N_{\boldsymbol{y}}} \sum_{i=1}^{N_{\boldsymbol{y}}} c_i$ is the aggregated uncertainty in generation $\boldsymbol{y}$, and $z = \mathbb{1}[\boldsymbol{y} = \boldsymbol{y}^*] \in \{0, 1\}$ is the correctness indicator. This formulation defines a convex optimization problem under the logistic loss, and ensures that the learned numerical scores yield effective verbal uncertainty.

### 4.3. Semantic Smoothing via Graph Laplacian Regularization

A key challenge in learning numerical scores for verbal uncertainty markers is data sparsity: some markers such as *"likely"* or *"possible"* appear frequently, while others like *"faint chance"* or *"virtually certain"* may occur rarely, making their learned values unstable. Intuitively, semantically similar markers should share similar uncertainty levels, unless strong evidence from data suggests otherwise.

To achieve that, we adopt graph Laplacian regularization to enforce smoothness by encouraging semantically similar verbal uncertainty markers to share consistent scores. This choice is consistent with established formulations in

graph-based learning, where the Laplacian energy is used to promote smoothness over similarity graphs, and with recent applications of semantic graph smoothing in NLP (Fu et al., 2022; Maskey et al., 2023; Fettal et al., 2024). Concretely, we construct a weighted similarity graph $G = (\mathcal{U}, E)$, where each edge weight $\boldsymbol{W}_{ij}$ captures the semantic similarity between markers $\boldsymbol{u}_i$ and $\boldsymbol{u}_j$, i.e., $\boldsymbol{W}_{ij} = s(\boldsymbol{u}_i, \boldsymbol{u}_j)$. By default, we use 3-gram Jaccard similarity as the semantic similarity measurement $s(\cdot, \cdot)$. Let $\boldsymbol{L} = \boldsymbol{D} - \boldsymbol{W}$ be the corresponding graph Laplacian, with $\boldsymbol{D}$ as the degree matrix. The semantic smoothing regularizer is then defined as

$$\mathcal{L}_{\text{lap}}(\mathbf{c}) = \gamma \, \mathbf{c}^\top \boldsymbol{L} \mathbf{c} = \gamma \sum_{i,j} \boldsymbol{W}_{ij}(c_i - c_j)^2.$$

where $\mathbf{c}$ denotes the vector of learnable numerical scores for all markers and $\gamma > 0$ is a hyperparameter controlling the regularization strength. This quadratic Dirichlet-energy penalty is the standard form for promoting smoothness on graphs; in the $p{=}2$ case used here, the Laplacian regularizer is a convex quadratic (Fu et al., 2022), while related variants such as fractional- and $p$-Laplacian formulations modulate the extent of smoothing and robustness (Fu et al., 2022; Maskey et al., 2023). By penalizing large discrepancies between semantically similar markers, this convex quadratic regularizer promotes smoother uncertainty assignments and leads to more robust verbal uncertainty, particularly for rare markers—empirically consistent with semantic graph smoothing on textual representations (Fettal et al., 2024).

The overall optimization objective is defined as the joint minimization of the BCE loss and the semantic smoothing regularizer, i.e., $\mathcal{L}(\boldsymbol{c}) + \mathcal{L}_{\text{lap}}(\boldsymbol{c})$. We utilize Adam to optimize our uncertainty scores. In Section 5.1, we provide detailed training protocols and hyperparameters. VOCAL constructs the UM-Lookup through a one-time optimization and can be directly applied to test-time generations for uncertainty quantification. Unlike logits- or sampling-based UQ methods, VOCAL does not require additional sampling or inference-time computation. In this way, VOCAL provides an efficient and effective approach for LLM uncertainty quantification.

## 5. Experiments

### 5.1. Experimental Setup

**Models.** Our evaluation is conducted on a set of state-of-the-art LLMs, including GPT-4o (Achiam et al., 2023), DeepSeek-V3.1 (DeepSeek-AI, 2024), GPT-3.5-Turbo (Brown et al., 2020), Qwen2.5-7B-Instruct (Qwen et al., 2025), Qwen2.5-72B-Instruct (Qwen et al., 2025), Llama-3.2-3B-Instruct and Meta-Llama-3.1-8B-Instruct (Grattafiori et al., 2024). To collect LLM generations for VOCAL, we adopt a verbal uncertainty prompting strategy (CoT with verbal uncertainty prompting). For other

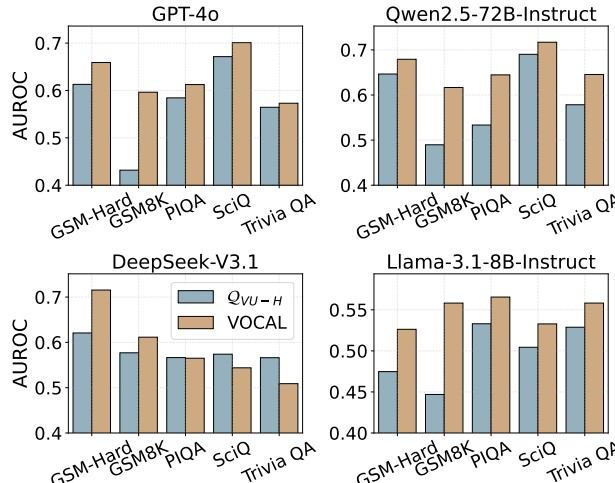

*Figure 4.* The evaluation results of VOCAL when comparing with human-sourced UM-Lookup, i.e., $\mathcal{Q}_{\text{VU-H}}$. It demonstrates that VOCAL produces LLM-tailored UM-Lookup table.

UQ baselines, we adopt the naive CoT prompt strategy for all the LLMs. Please refer to Section C for detailed prompt templates. A full specification of our generative configurations is provided in Section D.1.

**Datasets and Training Data Curation.** We consider 6 popular question-answering datasets: GSM-Hard (Gao et al., 2022), GSM8K (Cobbe et al., 2021), MedQA (Jin et al., 2020), PIQA (Bisk et al., 2020), SciQ (Welbl et al., 2017), and Trivia QA (Joshi et al., 2017). For a complete description of the datasets, please refer to Section D.2. We randomly select 300 questions from each dataset to curate the training set of VOCAL and randomly select 1,000 questions in the rest of each dataset. We will introduce the sample efficiency in this section.

**Hyperparameters.** By default, we set the graph Laplacian regularization strength to $\gamma = 5 \times 10^{-3}$ and use a learning rate of $1 \times 10^{-3}$. Training is conducted for up to 100 epochs with early stopping, where optimization terminates if the loss does not decrease within the most recent 10 epochs. In Section G, we provide the ablation study for $\gamma$.

**LLM UQ Baselines.** We consider popular logits- and sampling-based LLM UQ methods: Lexical Similarity (LS) (Fomicheva et al., 2020), Predictive Entropy (PE) (Malinin & Gales, 2020), Semantic Entropy (SE) (Kuhn et al., 2023a), Deg (Lin et al., 2023), sentSAR (Duan et al., 2024), G-NLL (Aichberger et al., 2025), and Semantic Density (SD) (Qiu & Miikkulainen, 2024). For sampling-based UQ baselines, we generate 5 samples for each question with a temperature of 0.8, and follow the default settings of these baselines.

**Evaluation Metrics.** Consistent with prior work (Kuhn et al., 2023a), we evaluate uncertainty quantification by measuring its ability to predict the correctness of a model's

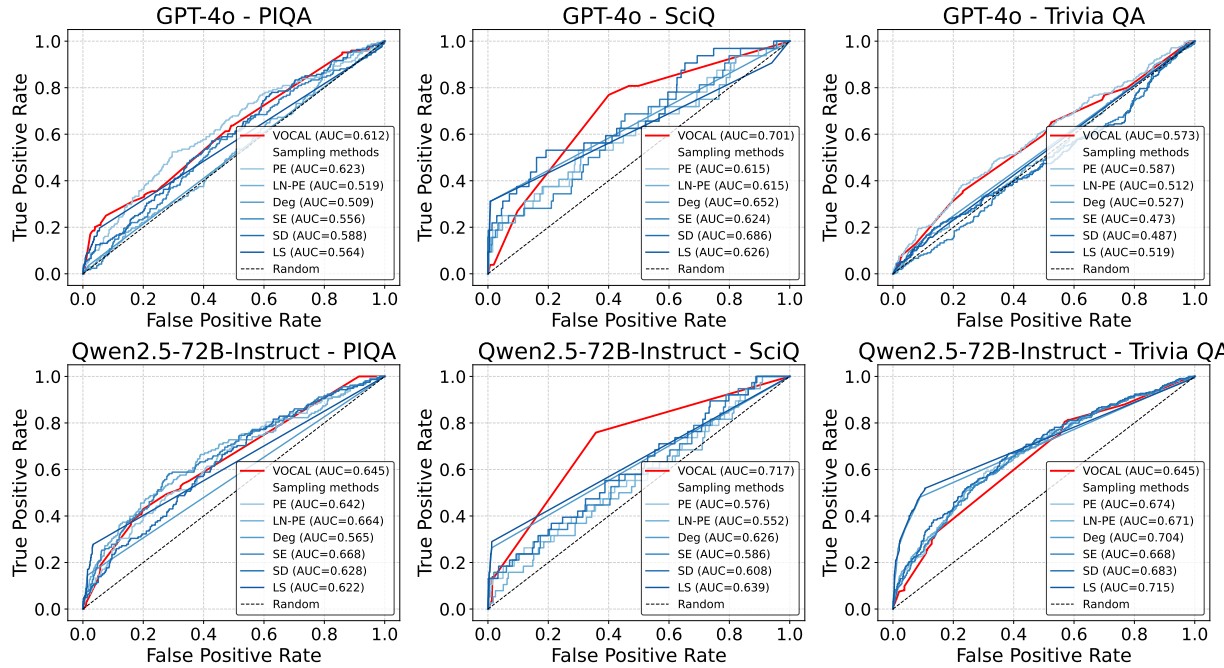

*Figure 5.* The evaluation results of VOCAL and multi-sample based UQ methods. It is worth noting that sampling-based methods rely on semantic consistency calculations, which are expensive and introduce latency in real-world deployment. It is shown that VOCAL achieves comparable performance to sampling-based UQ methods.

generated answers, using the Area Under the Receiver Operating Characteristic Curve (AUROC) as the evaluation metric.

**VOCAL is More Tailored for LLMs than Human-Sourced UM-Lookup.** As shown in Figure 4, VOCAL consistently improves AUROC over the human-sourced UM-Lookup ($\mathcal{Q}_{VU-H}$) across models and datasets, indicating a more effective mapping from linguistic uncertainty cues to correctness. The most representative gain appears on GSM8K, where the human lookup struggles: for GPT-4o, VOCAL boosts AUROC from 0.41 to around 0.6, demonstrating a substantial calibration improvement on math reasoning. Similar, though smaller, gains are observed on other benchmarks (e.g., GSM-Hard and PIQA/TriviaQA) for multiple models, suggesting that the benefit is not tied to a single dataset. Overall, the key conclusion is that uncertainty expressions are model-dependent, and learning an LLM-tailored uncertainty lookup (VOCAL) is more reliable than relying on a fixed, human-defined table, especially in failure modes where human heuristics do not align with the model's actual confidence behavior.

**VOCAL Outperforms 1-Sample UQ Methods.** As demonstrated in Table 1, VOCAL significantly outperforms single-sample UQ baselines such as G-NLL and Perplexity (PPL). Our method achieves the highest AUROC score in 5 out of the 6 evaluated settings. While PPL is marginally better on Trivia QA with GPT-4o, VOCAL's superiority is pronounced on more challenging reasoning datasets. For instance, on

*Table 1.* The comparison results between VOCAL and single-sample UQ baselines. It is shown that VOCAL is significantly better than these methods.

| Dataset | Model | G-NLL | PPL | VOCAL |
|---------|-------|-------|-----|-------|
| Trivia QA | GPT-4o | 0.538 | **0.575** | 0.573 |
|  | Qwen2.5-72B-Ins. | 0.627 | 0.619 | **0.645** |
| SciQ | GPT-4o | 0.663 | 0.648 | **0.700** |
|  | Qwen2.5-72B-Ins. | 0.568 | 0.555 | **0.717** |
| GSM-Hard | DeepSeek-V3.1 | 0.520 | 0.567 | **0.715** |
|  | Qwen2.5-72B-Ins | 0.507 | 0.580 | **0.679** |

GSM-Hard with DeepSeek-V3.1, VOCAL achieves an AUROC of 0.715, a substantial improvement over both G-NLL (0.520) and PPL (0.567). These results underscore the limitations of UQ methods that rely on a single greedy-decoded output and highlight the robustness of our approach.

**VOCAL is Comparable to Multi-Sampling UQ Methods.** Figure 5 further shows that VOCAL, despite using only a single response, can reach performance that is competitive with multi-sample UQ baselines that require multiple generations and expensive consistency computations. In particular, VOCAL often tracks the upper envelope of these sampling-based methods and can even be the best performer in some settings (e.g., on SciQ with Qwen2.5-72B, VOCAL achieves the top AUROC of 0.717). At the same time, we observe occasional gaps on certain datasets, such as TriviaQA, where several multi-sample metrics can surpass VOCAL. Overall,

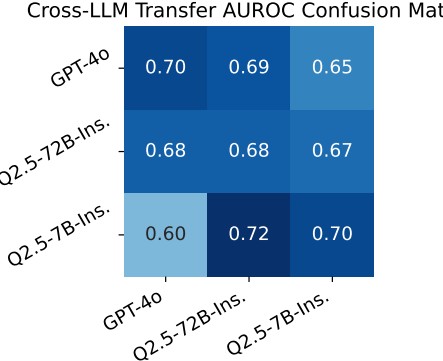

*Figure 6.* Cross-LLM transferability. LLMs share a substantial common structure in verbal uncertainty expression.

these results support the main conclusion: VOCAL offers a strong accuracy–efficiency trade-off, delivering near state-of-the-art uncertainty discrimination in many cases while avoiding the latency and cost of multi-sample semantic-consistency pipelines, which makes it more practical for real-world deployment.

**Calibration performance.** We evaluate the calibration quality of different uncertainty estimation methods using Expected Calibration Error (ECE) on GSM-Hard, GSM8K, and MedQA. For the vanilla setting, we min-max normalize the estimated uncertainty scores before computing ECE. We also apply two standard post-hoc calibration methods, temperature scaling (TS) and isotonic regression (IR), with all results obtained using 5-fold cross-validation. As shown in Table 3, VOCAL is competitive in the vanilla setting, achieving an average ECE of 0.1314, which is close to the best baseline. After calibration, VOCAL achieves stronger performance: VOCAL w/ TS obtains the lowest average ECE among all TS-calibrated methods, while VOCAL w/ IR further reduces the average ECE to 0.0281, achieving the best overall calibration performance. These results indicate that VOCAL provides a reliable uncertainty signal and can be effectively improved by standard post-hoc calibration methods, with IR yielding the strongest calibration performance.

**Number of Training Samples.** In Figure 9 (left), we find a strong positive correlation between the number of training samples and uncertainty quantification performance. Our results show that increasing the training data from 100 to 500 samples leads to a significant AUROC score improvement from approximately 0.52 to 0.60, demonstrating the benefit of a larger training set.

**Do LLMs Share Similar Confidence Level?** In Figure 6, we conduct cross-LLM transferability experiments by training and testing VOCAL on different LLM responses over the SciQ dataset. We follow the same training protocols (e.g., hyperparameters) as before. It is shown that the learned

*Table 2.* Mean probabilities of verbal uncertainty markers for GPT-4o and humans, sorted by the GPT-4o score. Row colors indicate the relationship between probabilities: uncolored for aligned values (within a 0.05 tolerance), and light gray where the GPT-4o probability is higher or the human probability is higher.

| Phrase | GPT-4o Prob. | Human Prob. |
|---|---|---|
| absolutely certain | 1.000 | 0.920 |
| confident | 0.839 | 0.900 |
| positive | 0.839 | 0.900 |
| sure | 0.839 | 0.830 |
| i think | 0.710 | 0.630 |
| almost certain | 0.677 | 0.920 |
| think | 0.645 | 0.490 |
| can | 0.355 | 0.570 |
| reasonable to assume | 0.355 | 0.605 |
| very likely | 0.355 | 0.853 |
| likely | 0.000 | 0.655 |

verbal-uncertainty indicators are largely transferable across models, with AUROC values remaining consistently in a competitive range (0.60–0.72). These results suggest that LLMs share a substantial common structure in verbal uncertainty expression, yet model-specific expression differences still prevent a fully universal, one-size-fits-all uncertainty mapping.

**How LLMs diverge from humans in verbal uncertainty quantification?** We compare our human-sourced UM-Lookup with a version optimized for GPT-4o on the SciQ dataset to analyze the alignment between human and LLM uncertainty expressions (Table 2). Our analysis reveals a significant divergence between the two, demonstrating that LLMs are not aligned with human verbal uncertainty. For instance, GPT-4o expresses 0.677 confidence for the phrase "*almost certain*", a term humans use with far more confidence (0.92), while conversely, it assigns a low probability to "*very likely*" (0.355), which humans rate with high confidence (0.853). This fundamental misalignment shows that human-derived tables are not directly transferable to LLMs, opening a new research direction into developing model-specific quantification methods like VOCAL.

## 6. Conclusion

This work investigates how LLMs diverge from humans in expressing verbal uncertainty. By constructing the first large-scale lookup table of human uncertainty markers and introducing VOCAL, an optimization-based alignment algorithm, we show that human-derived mappings only partially capture model behavior, while LLM-specific calibrations offer more reliable quantification. VOCAL achieves performance comparable to costly multi-sample UQ methods with much lower computational overhead. Our findings highlight

*Table 3.* Expected Calibration Error (ECE) of GPT-4o on GSM-Hard, GSM8K, and MedQA. Lower ECE indicates better calibration. For vanilla results, uncertainty scores are min-max normalized before computing ECE. Temperature scaling (TS) and isotonic regression (IR) are applied as post-hoc calibration methods.

| Method | Vanilla | | | | w/ TS | | | | w/ IR | | | |
|---|---|---|---|---|---|---|---|---|---|---|---|---|
| | GSM-Hard | GSM8K | MedQA | Avg | GSM-Hard | GSM8K | MedQA | Avg | GSM-Hard | GSM8K | MedQA | Avg |
| VOCAL | 0.2212 | 0.0462 | 0.1269 | 0.1314 | 0.0545 | **0.0197** | **0.0339** | **0.0360** | 0.0538 | **0.0051** | **0.0255** | **0.0281** |
| G-NLL | 0.0731 | 0.1476 | 0.1516 | 0.1241 | **0.0314** | 0.0623 | 0.0906 | 0.0614 | 0.0346 | 0.0298 | 0.0542 | 0.0395 |
| PPL | **0.0686** | 0.2553 | 0.1351 | 0.1530 | 0.0952 | 0.1316 | 0.0691 | 0.0986 | 0.0591 | 0.0286 | 0.0312 | 0.0396 |
| SE | 0.7100 | 0.5490 | 0.7568 | 0.6719 | 0.3964 | 0.4589 | 0.4468 | 0.4340 | 0.0591 | 0.0390 | 0.0512 | 0.0498 |
| PE | 0.2006 | 0.0598 | 0.1311 | 0.1305 | 0.0454 | 0.0290 | 0.0448 | 0.0397 | 0.0691 | 0.0364 | 0.0649 | 0.0568 |
| Deg | 0.2245 | 0.0482 | **0.1119** | **0.1282** | 0.1277 | 0.0228 | 0.0441 | 0.0649 | 0.0520 | 0.0258 | 0.0497 | 0.0425 |
| SD | 0.7091 | 0.5495 | 0.7576 | 0.6721 | 0.4006 | 0.4615 | 0.4493 | 0.4371 | **0.0449** | 0.0322 | 0.0310 | 0.0360 |

the importance of grounding LLM uncertainty in verbal expressions, offering both practical benefits for trustworthy deployment and new directions for human–AI alignment research.

## Limitations

Verbal uncertainty, while intuitive, faces several limitations. Its representation capacity is relatively weak, providing only coarse signals compared to probabilistic or semantic approaches. The extraction and cleaning of uncertainty markers also introduce challenges, as model outputs may contain ambiguous or overlapping expressions. Moreover, interpretations of verbal markers vary across domains and cultural contexts, limiting the generalizability of a single UM-Lookup. These issues highlight promising directions for future work on more expressive, robust, and context-aware verbal UQ methods.

## Impact Statement

This paper presents work whose goal is to advance the field of Machine Learning. There are many potential societal consequences of our work, none of which we feel must be specifically highlighted here.

## Acknowledgment

This work was partially supported by the Amazon Research Award and Cisco Faculty Award.

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

## A. Uncertainty Quantification in LLMs

For instance, from the Bayesian perspective, UQ can be derived by measuring the total uncertainty in the predictive distribution $p_{\boldsymbol{\theta}}(\boldsymbol{y} \mid \boldsymbol{x})$, where a common choice is the Predictive Entropy (PE) (Malinin & Gales, 2020), defined as

$$\mathcal{Q}_{\mathrm{PE}}(\boldsymbol{x}) = \int p_{\boldsymbol{\theta}}(\boldsymbol{y}|\boldsymbol{x}) \log(p_{\boldsymbol{\theta}}(\boldsymbol{y}|\boldsymbol{x})) \, d\boldsymbol{y} \approx -\frac{1}{N} \sum_i^N \log p_{\boldsymbol{\theta}}(\boldsymbol{y}^{(i)}|\boldsymbol{x}), \ \boldsymbol{y}^{(i)} \sim p_{\boldsymbol{\theta}}(\boldsymbol{y}|\boldsymbol{x}),$$

where $N$ is the number of samples and $p_{\boldsymbol{\theta}}(\boldsymbol{y}^{(i)}|\boldsymbol{x}) = \prod_i^{L_i} p_{\boldsymbol{\theta}}(z_i|z_{<i}, \boldsymbol{x})$ is the generative probability of $\boldsymbol{y}^{(i)}$ with length $L_i$. $z_i$ is the $i$-th token of $\boldsymbol{y}^{(i)}$. Moreover, Kuhn et al. (2023c) proposes Semantic Entropy (SE), which aggregates probability mass over semantic clusters of outputs:

$$\mathcal{Q}_{\mathrm{SE}}(\boldsymbol{x}) = -\frac{1}{C} \sum_i^C \log(p_{\boldsymbol{\theta}}(\boldsymbol{c}_i|\boldsymbol{x})), \ p_{\boldsymbol{\theta}}(\boldsymbol{c}_i|\boldsymbol{x}) = \sum_{\boldsymbol{y} \in \boldsymbol{c}_i} p_{\boldsymbol{\theta}}(\boldsymbol{y}|\boldsymbol{x}),$$

where $C$ is the number of semantic clusters and $\boldsymbol{c}_i$ is the $i$-th cluster consisting of generations $\boldsymbol{y}_i$ sharing the same semantics. These two examples illustrate how different realizations of $\mathcal{Q}$ target distinct aspects of output uncertainty.

## B. Human Verbal Uncertainty Expression

The lookup table presented below consolidates numerical probabilities for verbal uncertainty expressions (VUEs) from several key empirical studies. The aggregation process involved several steps to harmonize the data. For sources providing mean probability values, such as Lichtenstein & Newman (1967), the values were used directly (e.g., "likely" with mean=0.72). For studies reporting ranges, like Beyth-Marom (1982), we calculated the midpoint of the interquartile range to represent the central tendency (e.g., "likely" [0.55, 0.85] → 0.70). Data from Wesson & Pulford (2009), originally on a 1–7 point scale, was linearly rescaled to the probabilistic range [0, 1]. Meta-analytic estimates from Vogel et al. (2022) were incorporated to refine values and ensure cross-study consistency. The final probability for each VUE in Table 4 was derived by averaging these processed values, weighted by study prominence and term frequency where applicable. This table serves as the human-grounded benchmark for our analysis.

*Table 4.* Full Lookup Table for Verbalized Uncertainty Expressions (VUE) with their associated probabilities and frequencies.

| Uncertainty Expression | Uncertainty Probability | Frequency (N) |
| --- | --- | --- |
| Definite | 0.990 | 447.0 |
| Certain | 0.962 | 905.0 |
| Virtually certain | 0.950 | 447.0 |
| Almost certain | 0.920 | 782.0 |
| Absolutely certain | 0.920 | 96.0 |
| Very high chance | 0.915 | 27.0 |
| I know for a fact that it's... | 0.910 | 96.0 |
| I know it's... | 0.900 | 96.0 |
| Positive | 0.900 | 96.0 |
| Confident | 0.900 | 96.0 |
| Highly probable | 0.898 | 1081.0 |
| Nearly certain | 0.895 | 27.0 |
| No doubt | 0.870 | 96.0 |
| Very probable | 0.870 | 187.0 |
| Very likely | 0.853 | 1079.0 |
| Most likely | 0.850 | 27.0 |
| Close to certain | 0.835 | 27.0 |
| Sure | 0.830 | 96.0 |
| High chance | 0.810 | 27.0 |
| I have no doubt, I mean I'm sure it's... | 0.810 | 96.0 |
| Reasonably certain | 0.800 | 447.0 |
| Usually | 0.770 | 187.0 |
| Fairly confident | 0.760 | 96.0 |
| Reasonable assurance | 0.750 | 447.0 |
| Remember | 0.750 | 96.0 |
| Predictable | 0.740 | 146.0 |
| Good chance | 0.724 | 858.0 |
| Quite likely | 0.717 | 970.0 |
| Meaningful chance | 0.715 | 27.0 |
| Rather likely | 0.690 | 188.0 |
| Probable | 0.682 | 2311.0 |
| Believe | 0.670 | 96.0 |
| Pretty good chance | 0.670 | 188.0 |
| Fairly likely | 0.660 | 188.0 |
| Likely | 0.655 | 2227.0 |
| Suspect | 0.640 | 96.0 |
| I would say it's... | 0.640 | 96.0 |
| I could be mistaken but I'm sure it's... | 0.640 | 96.0 |
| I think it's... | 0.630 | 96.0 |
| Reasonable chance | 0.615 | 27.0 |
| One should assume | 0.610 | 27.0 |
| It seems to me | 0.605 | 27.0 |
| Reasonable to assume | 0.605 | 27.0 |
| Non-negligible chance | 0.600 | 27.0 |
| I'm not completely confident, but I think it's... | 0.600 | 96.0 |
| Quite probable | 0.600 | 447.0 |
| It seems | 0.590 | 27.0 |
| Somewhat likely | 0.590 | 187.0 |
| Rather | 0.580 | 124.0 |
| Better than even | 0.580 | 187.0 |

| Uncertainty Expression | Uncertainty Probability | Frequency (N) |
|---|---|---|
| I can't say for sure, but I think it's... | 0.570 | 96.0 |
| One can expect | 0.570 | 27.0 |
| I'm not certain, but it could be... | 0.560 | 96.0 |
| Slight odds in favor | 0.550 | 185.0 |
| I think it's.... but I can't be sure. | 0.550 | 96.0 |
| Slightly more than half the time | 0.550 | 188.0 |
| I guess it's... | 0.530 | 96.0 |
| I could be wrong, but I think it's... | 0.530 | 96.0 |
| I'm not sure, but it may be... | 0.530 | 96.0 |
| Possible (again?) | 0.520 | 447.0 |
| It's.... I think. | 0.520 | 96.0 |
| Fair chance | 0.510 | 188.0 |
| Tossup | 0.500 | 188.0 |
| Reasonably possible | 0.500 | 447.0 |
| It could be | 0.495 | 27.0 |
| May | 0.495 | 27.0 |
| Think | 0.490 | 96.0 |
| There is a chance | 0.485 | 27.0 |
| One must consider | 0.480 | 27.0 |
| Perhaps | 0.478 | 474.0 |
| Could be | 0.470 | 96.0 |
| Fighting chance | 0.470 | 186.0 |
| I think it's.... isn't it? | 0.470 | 96.0 |
| Possible | 0.464 | 2663.0 |
| Not inevitable | 0.455 | 27.0 |
| Maybe | 0.450 | 670.0 |
| Slight odds against | 0.450 | 185.0 |
| I'm guessing, but I would say it's... | 0.450 | 96.0 |
| Slightly less than half the time | 0.450 | 188.0 |
| Not quite even | 0.440 | 180.0 |
| Inconclusive | 0.430 | 153.0 |
| Don't know | 0.430 | 96.0 |
| Chance | 0.420 | 447.0 |
| Not sure | 0.420 | 96.0 |
| Not certain | 0.400 | 447.0 |
| Possibly | 0.380 | 447.0 |
| Can't rule out entirely | 0.365 | 27.0 |
| Uncertain | 0.356 | 1402.0 |
| Chances are not great | 0.345 | 27.0 |
| Somewhat unlikely | 0.310 | 186.0 |
| Somewhat doubtful | 0.300 | 447.0 |
| Small chance | 0.290 | 27.0 |
| Low chance | 0.280 | 27.0 |
| Fairly unlikely | 0.250 | 187.0 |
| Doubtful | 0.250 | 474.0 |
| Quite unlikely | 0.245 | 1193.0 |
| Rather unlikely | 0.225 | 374.0 |
| Not likely | 0.213 | 474.0 |
| Not very probable | 0.200 | 187.0 |
| Unlikely | 0.198 | 1752.0 |
| Not probable | 0.180 | 559.0 |
| Poor chance | 0.180 | 27.0 |

| Uncertainty Expression | Uncertainty Probability | Frequency (N) |
|---|---|---|
| Seldom | 0.160 | 188.0 |
| Not much chance | 0.160 | 186.0 |
| Improbable | 0.145 | 1081.0 |
| Very low chance | 0.140 | 27.0 |
| Barely possible | 0.130 | 180.0 |
| Faintly possible | 0.130 | 184.0 |
| Very unlikely | 0.116 | 1304.0 |
| Not possible | 0.100 | 559.0 |
| Almost impossible | 0.080 | 559.0 |
| Rare | 0.070 | 187.0 |
| Remote | 0.070 | 447.0 |
| Highly improbable | 0.052 | 851.0 |
| Impossible | 0.000 | 559.0 |

## C. Prompt LLMs to Express Verbal Uncertainty

This appendix details the two Chain-of-Thought (CoT) system prompts used in our experiments. The baseline **Standard CoT Prompt** requests a standard two-field JSON answer. In contrast, the **CoT with Verbal Uncertainty Prompt** extends this by requiring the model to incorporate UMs into its response and to report these expressions in an additional 'vue' field within a three-field JSON output.

---

**CoT Prompt**

You are a helpful and conversational AI assistant. Respond to questions in a natural, human-like tone.
Your response MUST be in valid JSON format with these two fields:

```
{
    "answer": "[Your conversational answer]",
    "final_answer": "[Your most specific answer]"
}
```

The "final_answer" should contain the most specific information possible, like a name, date, or place. The "answer" should be a natural explanation, as if you're talking to a friend.

---

**CoT with Verbal Uncertainty Prompt**

You are a knowledgeable and conversational AI assistant. Answer questions naturally with a human-like tone.
Your response should include:

1. A natural, conversational answer that incorporates verbalized uncertainty expressions (VUE) naturally within the text

2. A VUE section that lists all the uncertainty phrases you used in your answer

3. A final_answer section with the most specific answer you can provide

**IMPORTANT:** You MUST respond in valid JSON format with exactly these three fields:

```
{
    "answer": "[Your natural answer with embedded VUE expressions]",
    "vue": ["phrase1", "phrase2", "phrase3"],
    "final_answer": "[Your most specific answer]"
}
```

In your answer, naturally include uncertainty expressions including: {VUE_LIST: 'definite', 'certain', 'virtually certain', 'almost certain', ...}
Then in the vue field, provide an array of the uncertainty phrases you used. In the final_answer field, provide the most specific answer you can give (e.g., a name, place, date, etc.). Make your answer sound natural and conversational, as if explaining to a friend. Ensure your response is valid JSON that can be parsed.

# D. Experimental Settings

## D.1. Details of LLMs Generation

All models were queried using two distinct configurations. To assess correctness, we employed greedy decoding. To quantify uncertainty, we utilized multinomial sampling to draw 5 samples at a temperature of 0.8. All generated outputs were constrained by a maximum length of 512 tokens and a `top_p` value of 1.0.

## D.2. Datasets

**GSM8K** Cobbe et al. (2021) is a benchmark dataset featuring over 8,000 high-quality grade school math word problems. It is specifically designed to measure multi-step quantitative reasoning, with a key feature being that problems require several reasoning steps to solve. **GSM-Hard** Gao et al. (2022) is a challenging subset of GSM8K, curated to include only problems that necessitate the most complex and lengthy reasoning chains. **MedQA** Jin et al. (2020) is a large-scale multiple-choice dataset with over 11,000 questions derived from U.S. medical licensing exams, created to evaluate a model's capacity for deep medical knowledge. **PIQA** Bisk et al. (2020) is a commonsense reasoning benchmark containing over 18,000 examples in its training and validation sets. It is structured as a two-choice task that tests a model's understanding of physical interactions. **SciQ** Welbl et al. (2017) consists of approximately 13,700 crowdsourced science exam questions. Each question is multiple-choice and paired with a supporting text, testing both knowledge and comprehension. **TriviaQA** Joshi et al. (2017) is a high-quality reading comprehension dataset that contains over 650k question-answer pairs. Its distinct challenge lies in requiring models to find answers within large, unstructured evidence documents.

# E. Failure Case Analysis

In this section, we present failure cases of our method, including one where the model generates a correct answer with a low confidence score and another where it outputs an incorrect answer with a high confidence score. These analyses help clarify when verbal uncertainty fails and point toward possible directions for addressing these issues.

## E.1. Under-Confidence Despite Correctness

In the first case, the sampling-based baseline shows high confidence because the outputs across different generations are consistent and match the ground truth. In contrast, our method includes hedging expressions such as "not completely confident" or "I think," which drive the confidence score much lower, even though the final answer is correct.

---

Case 1: DeepSeek-V3.1 Model on Dataset PIQA

**Question:** How can I sneeze when I feel like I want to sneeze but can't?

**Choices:**
(1) Look directly in front of you this will cause a reaction in your body and you will sneeze.
(2) Look directly into sunlight, this will cause a reaction in your body and you will sneeze.

**Ground Truth Answer:** Look directly into sunlight, this will cause a reaction in your body and you will sneeze.

**Baseline Predictions:**
Answer: "If you're trying to trigger a sneeze, looking directly into sunlight often works because it stimulates the optic nerve and can cause a sneezing reflex."
Final Answer: Look directly into sunlight, this will cause a reaction in your body and you will sneeze.

Answer: "If you feel like you need to sneeze but can't, try looking into bright sunlight—it often triggers a sneeze reflex."
Final Answer: Look directly into sunlight, this will cause a reaction in your body and you will sneeze.

Answer: "If you're trying to trigger a sneeze, looking directly into sunlight can often help because it stimulates a reflex in some people."
Final Answer: Look directly into sunlight, this will cause a reaction in your body and you will sneeze.

Answer: "If you're trying to sneeze but can't, looking directly into sunlight often triggers a reaction that makes you sneeze."
Final Answer: Look directly into sunlight, this will cause a reaction in your body and you will sneeze.

Answer: "If you feel like you need to sneeze but can't, try looking into bright sunlight. This often triggers a photic sneeze reflex in many people."
Final Answer: Look directly into sunlight, this will cause a reaction in your body and you will sneeze.

**VUE Output:**
Answer: I'm *not completely confident*, but *I think* looking directly into sunlight is a common trick to trigger a sneeze.
Final Answer: Look directly into sunlight, this will cause a reaction in your body and you will sneeze.

VUE Phrases: ["not completely confident", "think"]

---

### E.2. Over-Confidence Despite Incorrectness

In the second case, the sampling-based baseline shows low confidence because the generated answers across different samples are inconsistent: some responses point to ectopic gastrin secretion, others to autoimmune destruction, and only one aligns with the ground truth. This disagreement shows the uncertainty when the model answers this question and aligns with the incorrect response. In contrast, our method uses high confidence expressions such as "fairly confident" or "most likely," but the final answer is incorrect.

---

**Case 2: DeepSeek-V3.1 Model on Dataset MedQA**

**Question:** An otherwise healthy 50-year-old man comes to the physician because of a 6-month history of increasingly frequent episodes of upper abdominal pain, nausea, vomiting, and diarrhea. He has had a 3.2-kg (7-lb) weight loss during this time. Physical examination shows bilateral pitting pedal edema. An endoscopy shows prominent rugae in the gastric fundus. Biopsy shows parietal cell atrophy. Which of the following is the most likely underlying cause?

**Choices:**
(1) Serotonin-secreting gastric tumor
(2) Proliferation of gastric mucus-producing cells
(3) Excessive somatostatin secretion
(4) Ectopic secretion of gastrin

**Ground Truth Answer:** Proliferation of gastric mucus-producing cells

**Baseline Predictions:**
Answer: "The findings—prominent gastric rugae, parietal cell atrophy, weight loss, and edema—point toward protein loss, likely due to gastric hypertrophy from ectopic gastrin secretion, as seen in Zollinger-Ellison syndrome."
Final Answer: Ectopic secretion of gastrin

Answer: "This sounds like a classic case of pernicious anemia with gastric parietal cell atrophy, which is often autoimmune-mediated. The prominent rugae, weight loss, and edema point toward protein loss and malabsorption, likely due to the lack of intrinsic factor from parietal cell damage."
Final Answer: Autoimmune destruction of parietal cells

Answer: "This sounds like a classic case of pernicious anemia due to autoimmune destruction of parietal cells, leading to atrophic gastritis and vitamin B12 deficiency. The prominent rugae, weight loss, and pedal edema are consistent with protein-losing enteropathy from the gastric atrophy."
Final Answer: Proliferation of gastric mucus-producing cells

Answer: "The symptoms and findings—like prominent rugae, parietal cell atrophy, weight loss, and edema—point toward a condition where there's excessive gastrin secretion, often due to a gastrinoma. This leads to high acid production, which can cause peptic ulcers and malabsorption issues, explaining the diarrhea and weight loss. The edema might be from protein loss or other complications. While other choices could affect the stomach, ectopic gastrin secretion fits best here."
Final Answer: Ectopic secretion of gastrin

Answer: "The symptoms and findings—like prominent rugae, parietal cell atrophy, weight loss, and edema—suggest a condition where there's excessive gastrin secretion, leading to gastric hyperplasia and malabsorption. This is most characteristic of Zollinger-Ellison syndrome, often caused by a gastrin-secreting tumor."
Final Answer: Ectopic secretion of gastrin

**VUE Output:**
Answer: I'm fairly confident this describes pernicious anemia with gastric atrophy, which is most likely caused by autoimmune destruction of parietal cells leading to intrinsic factor deficiency and vitamin B12 malabsorption.
Final Answer: Autoimmune destruction of parietal cells.

Phrases: ["fairly confident", "most likely"]

---

### E.3. Discussion.

These cases show that verbal expressions of uncertainty do not always align with a model's internal confidence. In some cases, hedging expression lowers the confidence even the predictions are correct. In other cases, the model conveys strong certainty while producing incorrect responses, which undermines trust and reliability. To address these challenges, future work should aim to capture uncertainty signals within specific domains and develop prompting strategies that foster clearer, more faithful representations of uncertainty.

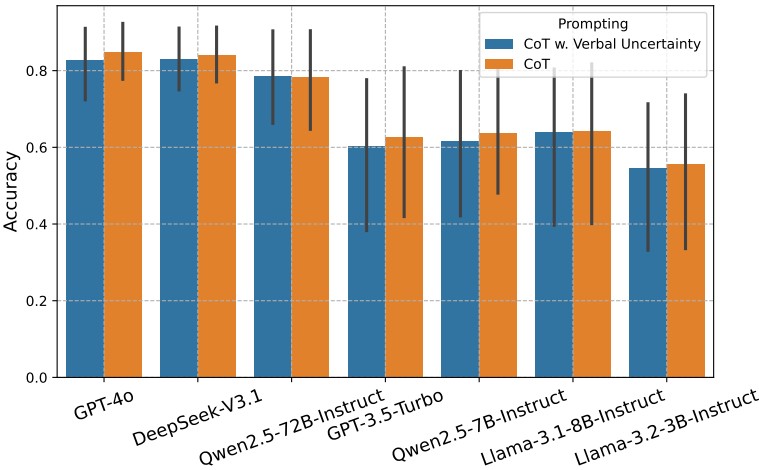

*Figure 7.* Verbal uncertainty prompting maintains general performance.

## F. Verbal Uncertainty Quantification with Human `UM-Lookup` Table

### F.1. Verbal Uncertainty Prompting Maintains General Performance

In Figure 7, we show that our verbal uncertainty prompting strategy does not significantly hurt the general performance of LLMs, which demonstrate the utility of `VOCAL` in applications.

### F.2. Advanced LLMs Express Diverse Uncertainty Markers

The UM distributions of each LLMs over all the datasets are presented in Figure 8.

## G. Optimized `UM-Lookup` Table

To complement our analysis, we provide optimized lookup tables that map verbal uncertainty markers to numerical uncertainty values. Specifically, Table 5 presents the optimized `UM-Lookup` for GPT-4o on the SciQ dataset. In addition, we report results for GPT-3.5-Turbo on MedQA (Table 6) and on SciQ (Table 7).

**Semantic smoothing** $\gamma$ **Ablation Study** Our analysis also reveals the model's sensitivity to the semantic smoothing hyperparameter, $\gamma$. The results indicate that performance is not monotonic with this value; the optimal AUROC is achieved at $\gamma = 0.005$, while lower or higher values lead to performance degradation, highlighting the importance of careful hyperparameter tuning.

## H. The Use of Large Language Models (LLMs)

For improved clarity and readability, we used OpenAI GPT-4o strictly as an editing aid. Its function was limited to correcting grammar, refining style, and polishing language, much like conventional grammar-checking tools or dictionaries. The model was not involved in generating scientific content or ideas, and its use remains in line with common standards for manuscript preparation.

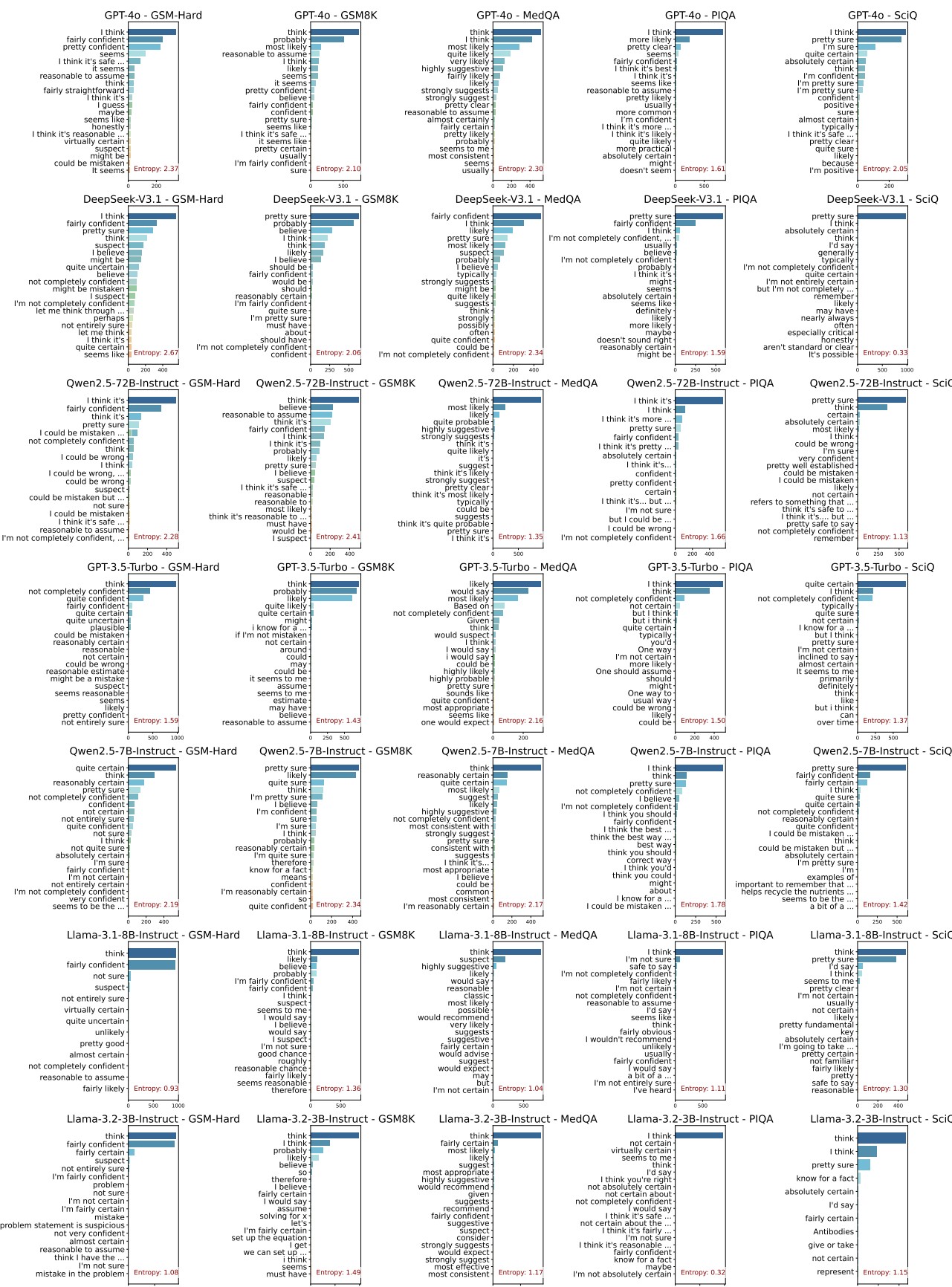

*Figure 8.* Verbal uncertainty marker distributions of LLMs.

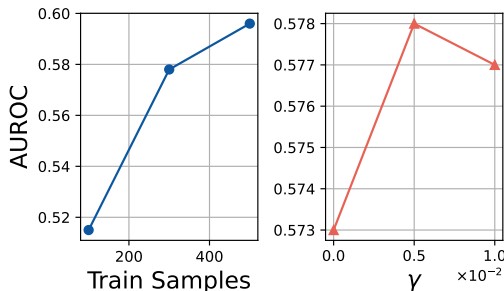

*Figure 9.* Ablation study on train samples and $\gamma$ measured by AUROC.

*Table 5.* Verbal uncertainty markers and their mean probabilities for GPT-4o on the SciQ dataset, sorted by probability.

| Phrase | Probability |
| --- | --- |
| absolutely certain | 1.000 |
| i'm sure | 1.000 |
| pretty sure | 1.000 |
| quite certain | 1.000 |
| confident | 0.839 |
| positive | 0.839 |
| sure | 0.839 |
| i'm pretty sure | 0.742 |
| i think | 0.710 |
| almost certain | 0.677 |
| i think it's safe to say | 0.677 |
| i'm confident | 0.645 |
| think | 0.645 |
| because | 0.355 |
| can | 0.355 |
| closely tied | 0.355 |
| pretty clear | 0.355 |
| quite similar | 0.355 |
| reasonable to assume | 0.355 |
| typically | 0.355 |
| very likely | 0.355 |
| likely | 0.000 |
| might have | 0.000 |

*Table 6.* Verbal uncertainty markers and their mean probabilities for GPT-3.5-Turbo on the MedQA dataset, sorted by probability.

| Phrase | Probability |
|---|---|
| best course of action | 1.000 |
| choice | 1.000 |
| given | 1.000 |
| highly probable | 1.000 |
| increased risk | 1.000 |
| most likely | 1.000 |
| pretty sure | 1.000 |
| quite confident | 1.000 |
| suggestive | 1.000 |
| based on | 0.999 |
| may be | 0.999 |
| may be needed | 0.999 |
| most appropriate | 0.999 |
| not definite | 0.999 |
| would expect | 0.999 |
| indication | 0.998 |
| most common | 0.998 |
| would most strongly | 0.998 |
| would suspect | 0.998 |
| likely | 0.997 |
| one would expect | 0.997 |
| should be | 0.997 |
| could be | 0.995 |
| i think | 0.908 |
| would say | 0.905 |
| seems | 0.739 |
| recommend | 0.506 |
| consider | 0.504 |
| seems like | 0.494 |
| i would say | 0.034 |
| would be | 0.013 |
| highly likely | 0.008 |
| sounds like | 0.004 |
| not completely confident | 0.003 |
| most concerning | 0.002 |
| indicating | 0.001 |
| likelihood | 0.001 |
| suspect | 0.001 |
| important | 0.000 |
| understandable | 0.000 |

*Table 7.* Verbal uncertainty markers and their mean probabilities for GPT-3.5-Turbo on the SciQ dataset, sorted by probability.

| Phrase | Probability |
| --- | --- |
| i know for a fact | 1.000 |
| pretty sure | 1.000 |
| but i think | 0.960 |
| inclined to say | 0.960 |
| not certain | 0.960 |
| quite certain | 0.920 |
| almost certain | 0.880 |
| definite | 0.880 |
| definitely | 0.880 |
| i know for a fact that it's... | 0.880 |
| like | 0.880 |
| primarily | 0.880 |
| not completely confident | 0.760 |
| can | 0.720 |
| i think | 0.720 |
| over time | 0.720 |
| quite sure | 0.560 |
| typically | 0.000 |

