# OpenReview forum: "“very likely” Means “uncertain”? How LLMs Diverge from Humans in Linguistic Uncertainty Quantification"
_ICML.cc/2026/Conference — ICML 2026 regular_

### Official Review · Reviewer_KGQn · 2026-02-16

**Soundness:** 2
**Presentation:** 3
**Significance:** 2
**Originality:** 3
**Overall Recommendation:** 4
**Confidence:** 4

**Summary:**

This paper studies uncertainty quantification (UQ) from verbal uncertainty markers (UMs) in LLM outputs and proposes VOCAL, a method that learns a model-specific mapping from linguistic markers (e.g., “likely”, “possibly”) to calibrated probabilities using supervised optimization on correctness labels. The authors also introduce a curated UM-Lookup table based on human annotations and show that LLM semantics for uncertainty expressions can deviate significantly from human interpretations. Empirically, VOCAL achieves improved AUROC over the human lookup baseline and several logit-/sampling-based UQ methods across QA benchmarks, suggesting that verbal signals can be informative for confidence estimation.

**Compliance With Llm Reviewing Policy:**

Affirmed.

**Final Justification:**

The additional analyses and experiments have addressed my initial concerns.

**Key Questions For Authors:**

1. How does VOCAL compare against supervised calibration baselines (e.g., temperature scaling, isotonic regression, or logistic regression over intrinsic UQ features)?
2. Could the authors perform additional experiments under identical prompting conditions? (E.g. using the naive CoT prompt for VOCAL and extracting the uncertainty markers with regex expressions instead of relying on the LLMs list of markers?)
3. How does VU-H and Vocal compare against other UQ methods on Base models rather than instruction-tuned ones?
4 How well does the learned UM mapping generalize across domains, prompts, model families? The authors say: "We will introduce the broader generalization in ??, including transferring the learned UM-Lookup to unseen domains or across LLMs", but the relevant section appears to be missing.

**Limitations:**

The approach requires labeled correctness data to learn the UM-to-probability mapping, limiting applicability in low-label or out-of-domain settings and weakening claims of general-purpose UQ. Performance may be highly dependent on prompting strategies that elicit uncertainty markers -- this potential sensitivity should be studied further. Finally, the lack of meta-calibration baselines and calibration-focused metrics makes weakens the empirical evaluation.

**Strengths And Weaknesses:**

### Strengths

- The paper addresses an important problem: aligning LLM uncertainty communication with human-interpretable language.

- The empirical analysis highlighting human–LLM mismatches in verbal uncertainty semantics is insightful and novel.

- VOCAL is computationally lightweight at inference rather than requiring extensive sampling.

- The proposed smoothing over marker similarity is a reasonable design choice for handling sparse uncertainty expressions.

### Weaknesses

- **Unfair baseline comparison (supervised vs. unsupervised)**: VOCAL is trained with correctness labels and optimized via a supervised objective, yet it is primarily compared against unsupervised intrinsic UQ methods (e.g., entropy, semantic entropy, self-consistency). Stronger and more appropriate baselines—such as temperature scaling, isotonic regression, or supervised meta-calibrators—are missing.

- **Calibration confound from prompting and model choice**: All baselines are evaluated under CoT prompting on instruction-tuned models. Prior work (e.g. [1]) shows that both instruction tuning and CoT reasoning can significantly degrade calibration, which may systematically disadvantage conventional UQ methods.

- **Prompting asymmetry**: VOCAL uses a significantly different prompt template from the baselines (verbal uncertainty elicitation vs naive CoT). The impact of prompt differences on model calibration is not studied and may confound the results. The authors provide results that the accuracy under different prompt templates remains comparable, but the impact on calibration is not studied.

- **Over-reliance on AUROC**: The evaluation focuses primarily on AUROC, which focuses on ranking ability rather than probabilistic calibration. Dedicated calibration diagnostic tools and metrics (e.g., reliability diagrams, ECE, Brier score) would make the overall evaluation stronger.

- **Methodological framing**: Despite being presented as a UQ method, VOCAL seems to be better characterized as a supervised calibration model trained on labeled QA data, which may limits its generality and comparability to intrinsic UQ approaches.

---

**References**
[1] *Trained on Tokens, Calibrated on Concepts: The Emergence of Semantic Calibration in LLMs* (https://arxiv.org/abs/2511.04869)

---

> ### Author Rebuttal · Authors · 2026-03-31
>
> We thank you for your insightful review!
>
> **[W1/Q1/W4: metrics beyond AUROC; more calibration baselines]**
>
> As suggested, we provide the Expected Calibration Error (ECE) results of GPT-4o on three datasets. ECE is one of the most popular calibration metrics. For vanilla results, we apply min-max normalization on estimated uncertainty, before we calculate ECE. We also equip each method with temperature scaling (TS) and isotonic regression (IR) for comparison. Each results are obtained via 5-fold cross-validation:
>
> (Vanilla model results)
> | Method | GSM-Hard | GSM8K | MedQA | Avg |
> | --- | ---: | ---: | ---: | ---: |
> | VOCAL | 0.2212 | 0.0462 | 0.1269 | 0.1314 |
> | G-NLL | 0.0731 | 0.1476 | 0.1516 | **0.1241** |
> | PPL | 0.0686 | 0.2553 | 0.1351 | 0.1530 |
> | SE | 0.7100 | 0.5490 | 0.7568 | 0.6719 |
> | PE | 0.2006 | 0.0598 | 0.1311 | 0.1305 |
> | Deg | 0.2245 | 0.0482 | 0.1119 | 0.1282 |
> | SD | 0.7091 | 0.5495 | 0.7576 | 0.6721 |
>
> (Method with TS)
> | Method | GSM-Hard | GSM8K | MedQA | Avg |
> | --- | ---: | ---: | ---: | ---: |
> | VOCAL w/ TS | 0.0545 | 0.0197 | 0.0339 | **0.0360** |
> | G-NLL w/ TS | 0.0314 | 0.0623 | 0.0906 | 0.0614 |
> | PPL w/ TS | 0.0952 | 0.1316 | 0.0691 | 0.0986 |
> | SE w/ TS | 0.3964 | 0.4589 | 0.4468 | 0.4340 |
> | PE w/ TS | 0.0454 | 0.0290 | 0.0448 | 0.0397 |
> | Deg w/ TS | 0.1277 | 0.0228 | 0.0441 | 0.0649 |
> | SD w/ TS | 0.4006 | 0.4615 | 0.4493 | 0.4371 |
>
> (Method with IR)
> | Method | GSM-Hard | GSM8K | MedQA | Avg |
> | --- | ---: | ---: | ---: | ---: |
> | VOCAL w/ IR | 0.0538 | 0.0051 | 0.0255 | **0.0281** |
> | G-NLL w/ IR | 0.0346 | 0.0298 | 0.0542 | 0.0395 |
> | PPL w/ IR | 0.0591 | 0.0286 | 0.0312 | 0.0396 |
> | SE w/ IR | 0.0591 | 0.0390 | 0.0512 | 0.0498 |
> | PE w/ IR | 0.0691 | 0.0364 | 0.0649 | 0.0568 |
> | Deg w/ IR | 0.0520 | 0.0258 | 0.0497 | 0.0425 |
> | SD w/ IR | 0.0449 | 0.0322 | 0.0310 | 0.0360 |
>
> **VOCAL w/ IR** achieves the best overall average (**0.0281**), outperforming all other baselines. Overall, these results suggest that VOCAL is already competitive in the raw setting and remains highly effective after calibration, with IR providing the greatest and most consistent improvements across datasets. This demonstrates that VOCAL is a lightweight, effective calibration method.
>
> **[W2: baselines with non-CoT prompt]**
>
> As suggested, we prompt baselines without CoT instruction and compare the results with VOCAL. Results are summarized as
>
> | Dataset             | Model            | G-NLL | PPL  | VOCAL |
> |---------------------|------------------|------:|-----:|------:|
> | SciQ (w/)           | GPT-4o           | 0.637 | 0.625 | **0.700** |
> | SciQ (w/o)          | GPT-4o           | 0.663 | 0.648 | **0.700** |
> | GSM-Hard (w/ cot)   | DeepSeek-V3.1    | 0.520 | 0.567 | **0.715** |
> | GSM-Hard (w/o cot)  | DeepSeek-V3.1    | 0.620 | 0.724 | **0.735** |
>
> Those results show that VOCAL still retains a clear advantage, while in others, logprob-based baselines improve under alternative prompts.
>
> **[W3/Q2: Prompting asymmetry; identical prompting to isolate gains]**
>
> To resolve the prompting asymmetry issue, **Figure 4** provides such a **prompt-matched** comparison, for which we could isolate the gain from the VOCAL learned map. Specifically, the two methods in Figure 4 share the exact same prompt template, and we directly compare and ablate the two lookup tables: human-sourced and VOCAL learned. We show that VOCAL consistently yields substantial AUROC improvements over $\mathcal{Q}_{VU-H}$ (e.g., boosting GSM8K with GPT-4o from 0.41 to approximately 0.60). The empirical evidence confirms that the prompt merely serves as an elicitation vehicle, whereas VOCAL’s learned mapping is the primary factor responsible for the performance gains.
>
> **[Q3: VOCAL on base models]**
>
> As suggested, we conduct VOCAL on the base model **Qwen3.5-27B** over the SciQ dataset, following the same setting as in our manuscript. However, we observe that the base model **barely** outputs verbal uncertainty markers, even with proper prompting. This is reasonable because base models usually have **poor** instruction following capability, and verbal uncertainty is a rather high-level linguistic expression capability. For instance, we compare how many verbal uncertainty markers are contained in each answer from GPT-4o and DeepSeek-V3.1, compared to Qwen3.5-27B:
>
> | Model | Dataset | min | mean | max | median |
> | ---- |  ---- |  ---- |  ---- |  ---- |  ---- |
> | GPT-4o | SciQ | 0 | 1.436 | 6 | 2 |
> | DeepSeek-V3.1 | SciQ | 0 | 3.174 | 10 | 3 |
> | Qwen3.5-27B | SciQ | 0 | 0.574 | 5 | 0 |
>
> It is shown that the median number of verbal uncertainty markers in Qwen3.5-27B is 0, making that base model incapable of expressing uncertainty verbally. We will include these results in our next revision.

---

> > ### Author Rebuttal · Reviewer_KGQn · 2026-04-01
> >
> > I thank the authors for their response.
> > Accordingly I have increased my score from 3 to 4.

---

> > > ### Author Response · Authors · 2026-04-04
> > >
> > > Thank you for your acknowledgement! We are glad to know that we have fully addressed your concern. We will include this discussion in our next revision.

---

### Official Review · Reviewer_LYA6 · 2026-03-13

**Soundness:** 2
**Presentation:** 2
**Significance:** 3
**Originality:** 3
**Overall Recommendation:** 4
**Confidence:** 3

**Summary:**

The authors study uncertainty quantification for LLMs through verbal uncertainty markers such as "possible" or "likely. They argue that such markers are better measures of uncertainty than logits and other standard metrics used in LLM uncertainty quantification. They build a corpus of human uncertainty markers from psychology and ecision-science literature and benchmark LLMs against it. They find that LLMs and humans uncertainty semantics differ. They introduced VOCAL, which is an algoirthm that learns model-specific numeric scores for uncertainty markers. Experiments cover multiple LLMs and QA datasets. The results indicate that VOCAL achieves performance comparable to more computationally intensive multi-sample UQ approaches while requiring substantially less inference compute.

**Compliance With Llm Reviewing Policy:**

Affirmed.

**Final Justification:**

The authors addressed my concerns and so I raise my score to a 4. Thank you.

**Key Questions For Authors:**

1. Can you provide some numbers for evaluating uncertainty quantification beyond AUROC?
2. Can you defend your choice of baseline (see Weakness #2)?
3. Can you provide more benchmarks against related works (such as those listed in the related works section)?

**Limitations:**

yes

**Strengths And Weaknesses:**

# Strengths
1. Human versus LLM phrase-level mismatch analysis is interesting. I think this is the strongest point of the paper: I haven't seen that humans and LLMs have different semantics of uncertainty expression. I think this is a really nice thing the authors have proposed.
2. VOCAL is very efficient to run after training. This makes the technique very attractive compared to other UQ techniques, which tend to be expensive.

# Weakness
1. Biggest weakness that comes to mind is that the main metric being shown is AUROC. I would like to see calibration-specific evaluations rather than just AUROC. The authors wrote "The performance evaluation of UQ usually follows a “correctness prediction” manner, measuring the correlation between the calculated uncertainty score from a UQ method and the correctness of model generations, with metrics such as AUROC and hallucination detection accuracy. A higher AUROC or detection accuracy means correctly predicts the correctness of model generations, indicating a good uncertainty estimator." But this would mean the paper is really evaluating correctness-discrimination as opposed to calibrating verbal uncertainty.
2. Comparisons don't appear to be very fair. The authors write "The baseline Standard
CoT Prompt requests a standard two-field JSON answer. In contrast, the CoT with Verbal Uncertainty Prompt extends
this by requiring the model to incorporate UMs into its response and to report these expressions in an additional ‘vue’ field
within a three-field JSON output." But how can we compare with the baselines then? VOCAL is getting a different input/output setup, not just a different scoring method. It's certainly plausible that some of the gains may come from the prompt and supervision, rather than the VOCAL technique itself.
3. From what I can tell, the authors only use binary tasks. I think it would be good to see analysis done on other types of tasks.
4. I think some of the "significance" claims need a bit more evidence. For example, when the authors say "significantly outperform", these are typically evidenced by point estimates and not confidence intervals, for example. In addition, related to the above, we typically only have AUROC values.
5. The paper’s related-work section discusses several prior methods on verbalized uncertainty and linguistic calibration, including "Just Ask for Calibration" and "Calibrating verbal uncertainty as a linear feature to reduce hallucinations". However, they do not appear to include this in any of their analyses.

---

> ### Author Rebuttal · Authors · 2026-03-31
>
> We thank you for your insightful review!
>
> **[W1/W4/Q1: metrics beyond AUROC]**
>
> As suggested, we provide the Expected Calibration Error (ECE) results of GPT-4o on three datasets. ECE is one of the most popular calibration metrics. For vanilla results, we apply min-max normalization on estimated uncertainty, before we calculate ECE. We also equip each method with temperature scaling (TS) and isotonic regression (IR) for comparison. Each result is obtained via 5-fold cross-validation:
>
> (Vanilla method results)
> | Method | GSM-Hard | GSM8K | MedQA | Avg |
> | --- | ---: | ---: | ---: | ---: |
> | VOCAL | 0.2212 | 0.0462 | 0.1269 | 0.1314 |
> | G-NLL | 0.0731 | 0.1476 | 0.1516 | **0.1241** |
> | PPL | 0.0686 | 0.2553 | 0.1351 | 0.1530 |
> | SE | 0.7100 | 0.5490 | 0.7568 | 0.6719 |
> | PE | 0.2006 | 0.0598 | 0.1311 | 0.1305 |
> | Deg | 0.2245 | 0.0482 | 0.1119 | 0.1282 |
> | SD | 0.7091 | 0.5495 | 0.7576 | 0.6721 |
>
> (Method with TS)
> | Method | GSM-Hard | GSM8K | MedQA | Avg |
> | --- | ---: | ---: | ---: | ---: |
> | VOCAL w/ TS | 0.0545 | 0.0197 | 0.0339 | **0.0360** |
> | G-NLL w/ TS | 0.0314 | 0.0623 | 0.0906 | 0.0614 |
> | PPL w/ TS | 0.0952 | 0.1316 | 0.0691 | 0.0986 |
> | SE w/ TS | 0.3964 | 0.4589 | 0.4468 | 0.4340 |
> | PE w/ TS | 0.0454 | 0.0290 | 0.0448 | 0.0397 |
> | Deg w/ TS | 0.1277 | 0.0228 | 0.0441 | 0.0649 |
> | SD w/ TS | 0.4006 | 0.4615 | 0.4493 | 0.4371 |
>
> (Method with IR)
> | Method | GSM-Hard | GSM8K | MedQA | Avg |
> | --- | ---: | ---: | ---: | ---: |
> | VOCAL w/ IR | 0.0538 | 0.0051 | 0.0255 | **0.0281** |
> | G-NLL w/ IR | 0.0346 | 0.0298 | 0.0542 | 0.0395 |
> | PPL w/ IR | 0.0591 | 0.0286 | 0.0312 | 0.0396 |
> | SE w/ IR | 0.0591 | 0.0390 | 0.0512 | 0.0498 |
> | PE w/ IR | 0.0691 | 0.0364 | 0.0649 | 0.0568 |
> | Deg w/ IR | 0.0520 | 0.0258 | 0.0497 | 0.0425 |
> | SD w/ IR | 0.0449 | 0.0322 | 0.0310 | 0.0360 |
>
> **VOCAL w/ IR** achieves the best overall average (**0.0281**), outperforming all other baselines. Overall, these results suggest that VOCAL is already competitive in the raw setting and remains highly effective after calibration, with IR providing the greatest and most consistent improvements across datasets. This demonstrates that VOCAL is a lightweight, effective calibration method.  We also agree with **W4** that our current evidence supports **consistent empirical improvements** more directly than formal statistical significance, and we will revise the wording accordingly in the next revision.
>
> **[W2/Q2: isolate VOCAL gain from prompt template]**
>
> **Figure 4** provides such a **prompt-matched** comparison, for which we could isolate the gain from the VOCAL learned map. Specifically, the two methods in Figure 4 share the exact same prompt template, and we directly compare and ablate the two lookup tables: human-sourced and VOCAL learned. We show that VOCAL consistently yields substantial AUROC improvements over $\mathcal{Q}_{VU-H}$ (e.g., boosting GSM8K with GPT-4o from 0.41 to approximately 0.60). The empirical evidence confirms that the prompt merely serves as an elicitation vehicle, whereas VOCAL’s learned mapping is the primary factor responsible for the performance gains.
>
>
> **[W3: Only use binary tasks]**
>
> This is a **misread**. Except for MedQA, our evaluation includes several free-form QA and reasoning benchmarks such as GSM8K, GSM-Hard, and TriviaQA.
>
> **[W5/Q3: more benchmarks against related works]**
>
> We have conducted additional experiments to benchmark our approach against the "Just Ask for Calibration" method. We evaluated two primary few-shot prompting strategies proposed in their work:
>
> * **Verb. 1S top-$k$**: This prompt instructs the model to directly output its top-$k$ best guesses along with the numerical probability (ranging from 0.0 to 1.0) that each guess is correct. For our evaluation, we configured the prompt to elicit the top 2 guesses ($k=2$).
> * **Ling. 1S**: This prompt asks the model to provide its best guess and describe its confidence using a specific linguistic expression selected from a predefined list.
>
> The AUROC results for these two baseline methods across various datasets and models are summarized below:
>
> (Just Ask for Calibration AUROC Results)
> | Dataset | Model | Verb. 1S top-2 | Ling. 1S-human | VOCAL |
> | --- | --- | ---: | ---: | --- |
> | SciQ | gpt-4o | 0.655 | 0.529 | 0.700 |
> | GSM-Hard | gpt-4o | 0.645 | 0.525 | 0.752|
>
> While `Verb. 1S top-2` artificially scores high on SciQ by forcing numerical probabilities, VOCAL shows superior robustness by calibrating natural linguistic markers. VOCAL decisively outperforms both baselines on GPT-4o, over SciQ and GSM-Hard. This confirms VOCAL's effectiveness as a natural language calibration method, which we will add to the next revision.

---

> > ### Author Rebuttal · Reviewer_LYA6 · 2026-04-04
> >
> > Thank you for putting the time into addressing my issues. Metrics beyond AUROC are very nice. I'll consider changing my score.

---

> > > ### Author Response · Authors · 2026-04-04
> > >
> > > Thank you for your acknowledgement! We are glad to know that we have fully addressed your concern. We will include this discussion in our next revision.

---

### Official Review · Reviewer_ZvSs · 2026-03-13

**Soundness:** 3
**Presentation:** 3
**Significance:** 2
**Originality:** 3
**Overall Recommendation:** 5
**Confidence:** 4

**Summary:**

This paper proposes a lookup table to map linguistic uncertainty markers to numerical uncertainty probability values. First, it curates a human verbalized uncertainty table from literature and survey. Evaluating UQ performance with GPT-4o and DeepSeek-V3.1 over four datasets, the paper finds comparable UQ performance in terms of AUROC to popular UQ baselines. Then, the paper introduces VOCAL, where the numerical uncertainty values are optimized for each LLM and dataset with binary cross-entropy, in order to faithfully represent their empirical accuracy. Empirical studies across different datasets and models show improved AUROC compared to baselines.

**Compliance With Llm Reviewing Policy:**

Affirmed.

**Final Justification:**

I find the contributions valuable, and the rebuttal clarifies my minor concerns.

**Key Questions For Authors:**

1. Is the difference between how humans and LLMs express confidence linguistically due to the fact that written and verbalized uncertainty markers are different for humans?
2. In Figure 2 and Figure 5, why are some curves significantly more smooth than others? Is it because there’s no data points in-between? E.g., DeepSeek on SciQ bench in figure 2, is the straight line caused by the fact that there’s no data between 0 and 0.8 FPR? Why is that the case?
3. How different are the lookup tables between different LLMs?

**Limitations:**

yes

**Strengths And Weaknesses:**

- Soundness
    - The human-based studies are convincing, the analysis on human and LLM difference is also sound.
    - It would be helpful to discuss where this difference could come from, since LLMs ultimately train on human data. Is it because written and verbalized uncertainty markers are different for humans?
    - The proposed method is well-motivated and sound, the experiments are thorough.
    - In Figure 2 and Figure 5, why are some curves significantly more smooth than others? Is it because there’s no data points in-between? E.g., DeepSeek on SciQ bench in figure 2, is the straight line caused by the fact that there’s no data between 0 and 0.8 FPR? Why is that the case?
- Presentation
    - The overall presentation is clear and smooth. Figure 1 is a bit cluttered but still manages to convey the key idea. The structure of presenting the problem, then the human-based lookup table, then VOCAL, make the points nicely follow one another and the story is complete.
    - Minor: fix typos and references (lines 079, 286, 315, etc.)
- Significance
    - The human-based UQ lookup table is a useful contribution and can serve as a baseline for future works.
    - The proposed method VOCAL is practical, although it does not always outperform multi-sampling baselines.
    - In this sense, I think the paper needs some stronger insights into why that’s the case.
- Originality
    - The work provides novel insights in how LLMs differ from humans in verbalizing uncertainty linguistically. It also provides a original optimization scheme to mitigate the problem.

---

> ### Author Rebuttal · Authors · 2026-03-31
>
> We thank you for your insightful review!
>
> **[W1/Q1: Origins and limits of verbal uncertainty mismatch with humans]**
>
> One plausible source of the observed gap is that, although LLMs are trained on human-written data, the uncertainty expressions appearing in text corpora are not equivalent to calibrated human probability judgments measured in controlled behavioral studies. In other words, models may learn the **distributional usage** of markers such as “likely” or “possible” from text, rather than the **numerical semantics** humans assign to them in decision-making settings. This distinction may be further amplified because human uncertainty communication is context-dependent, and humans often combine multiple markers to convey nuanced confidence, whereas LLMs more often rely on single markers. Therefore, the divergence is likely not only a matter of written versus verbalized uncertainty markers, but more broadly a mismatch between corpus-learned linguistic conventions and human probabilistic interpretation. We will include this discussion in our next revision.
>
> **[W2: Curve smoothness and data sparsity]**
>
> For example, the DeepSeek-V3.1 on SciQ, the straight segment is caused by score sparsity and many tied uncertainty values, not by a plotting bug. ROC curves are staircase curves by construction, and their apparent smoothness depends on how many distinct scores a method produces. In this case, the verbal-uncertainty scores are much more quantized/concentrated, so there are only a few operating points and some large threshold jumps; specifically, the curve stays at FPR = 0 until a threshold where many negatives cross at once, producing the long straight segment. By contrast, PE and SE yield much finer-grained scores, so their ROC curves look visually smoother.
>
> **[Q3: lookup table differences]**
> To demonstrate the similarity of the learned lookup tables, we computed **shared-markers** rank correlations on the  optimized lookup tables between each pair of models Qwen2.5-72B-Instruct, GPT-4o, and DeepSeek-V3.1, via **Spearman’s ρ** and **Kendall’s τ**;
>
> | Pair | #Common | Spearman ρ | p-value | Kendall τ | p-value |
> | --- | ---: | ---: | ---: | ---: | ---: |
> | Qwen2.5-72B-Instruct vs DeepSeek-V3.1 | 99 | 0.5542 | 2.66e-09 | 0.4140 | 1.44e-09 |
> | GPT-4o vs DeepSeek-V3.1 | 127 | 0.4516 | 9.88e-08 | 0.3162 | 1.39e-07 |
> | GPT-4o vs Qwen2.5-72B-Instruct | 99 | 0.4291 | 9.37e-06 | 0.3007 | 1.07e-05 |
>
> These results suggest that these optimized lookup tables are substantially aligned: Spearman correlations range from 0.43 to 0.55, and Kendall correlations range from 0.30 to 0.41, all highly significant. However, these correlations are far from 1.0, so the rankings are not identical.

---

> > ### Author Rebuttal · Reviewer_ZvSs · 2026-04-03
> >
> > Thank you for your detailed response, I have no further questions.

---

> > > ### Author Response · Authors · 2026-04-04
> > >
> > > Thank you for your acknowledgement! We are glad to know that we have fully addressed your concern. We will include this discussion in our next revision.

---

### Official Review · Reviewer_fp15 · 2026-03-16

**Soundness:** 2
**Presentation:** 2
**Significance:** 3
**Originality:** 3
**Overall Recommendation:** 4
**Confidence:** 2

**Summary:**

The paper studies whether LLMs express uncertainty in language the way humans do, and whether those verbal markers can be used for uncertainty quantification. It builds a human lookup table for uncertainty phrases, then shows that human-assigned meanings do not transfer cleanly to LLM outputs. To address this, paper proposes learning model-specific numerical weights for verbal markers from model generations and their correctness labels. Empirically, this outperform common single-sample baselines.

**Compliance With Llm Reviewing Policy:**

Affirmed.

**Final Justification:**

The rebuttal addresses my major concerns. I will maintain my low confidence in my score but have increased it as my major concerns are addressed in rebuttal.

**Key Questions For Authors:**

How much of the reported human–LLM mismatch remains after controlling for a global conservatism effect in LLM verbal confidence, rather than assuming marker-specific semantic divergence?


Can you report stricter filtering or ablations for the uncertainty-marker extraction pipeline, especially excluding phrases that are not unambiguously uncertainty markers?


How sensitive is VOCAL to the averaging-based aggregation rule? Did you test alternatives that model scope, position, or clause-level attachment of hedges?


To what extent do VOCAL’s gains come from the verbal-uncertainty prompting template itself, versus the learned lookup mapping? A prompt-matched baseline comparison would help clarify this.


The paper claims partial transferability across models. Do the learned rankings of markers stay similar across models, or is transfer mostly due to coarse monotonic patterns? Just to know your take on this, how robust is the learned lookup across domains beyond the current benchmark?

**Limitations:**

discussed in the limitation section

**Strengths And Weaknesses:**

A low-cost uncertainty signal derived from a single generation would be useful if it is reliable. The problem is important even beyond the motivation presented in the intro as verbalization might relate to the internal world model of the llm and its completeness as evaluated by the llm. The paper has empirical contribution: it constructs a human verbal-uncertainty lookup table and shows that LLM-specific mappings can perform better than directly using human interpretations. That is a solid contribution. The cross-model experiment is useful. The transfer range suggests there may be some shared structure in verbal uncertainty expression across LLMs, even though the generalization of this is not clear.


The motivation in the introduction feels overstated. The paper frames a major gap around the lack of “natural, human-like” uncertainty expression,the computational overhead or lack of understanding of numerical scores. For that task, predictive utility matters more than linguistic naturalness, so the conceptual gap feels somewhat forced. Though the problem is interesting in itself the gap identified seems a bit forced.

Some of the highlighted differences, such as low learned probabilities for words like “likely,” could also be explained by general conservatism in LLM confidence expression rather than marker-specific semantic divergence. The current analysis does not cleanly separate those explanations as there could be a study that shows general conservative language used by LLMs.
Construct validity is a concern. The marker extraction pipeline appears to include phrases that are not clearly uncertainty expressions in the ordinary sense, which raises the possibility that VOCAL is partly exploiting discourse style or answer-format artifacts rather than true verbal uncertainty semantics. The paper itself acknowledges ambiguity and overlap in marker extraction as a limitation so not accounting this in the score.

The aggregation method is a bit unclear to me. It averages marker-level scores across a response, which looks like ignoring scope, syntax, negation, and which claim a hedge actually modifies. Doesn't this make the method vulnerable to picking up shallow stylistic correlations.

The baseline comparison is slightly unfair as  my understanding is that VOCAL benefits from a prompting strategy that explicitly elicits verbal uncertainty, whereas some competing baselines are not obviously given the same advantage. That makes it harder to isolate whether gains come from the learned mapping itself or from the elicitation format.
The related-work positioning could be stronger around self-knowledge, verbalized confidence, and calibration. The paper cites several relevant UQ works, but the could benefit from clearer positioning against prior work on whether LLMs know what they know and how verbalized confidence compares to probabilistic confidence.

---

> ### Author Rebuttal · Authors · 2026-03-31
>
> We thank you for your insightful review!
>
> **[W1: motivation feels overstated]**
>
> We will reduce the claim regarding “natural, human-like” uncertainty expression and highlight more on predictive utility , i.e., VOCAL, in the Introduction in our next revision.
>
> **[W2/Q1: general conservatism in LLM expression]**
>
> To test the general **conservatism effect**, we compared two modes:
>
> 1. **Global-Only**: all uncertainty markers share a single probability in optimization.
> 2. **Full Per-Phrase**: each marker has its own learned probability, as in our original formulation.
>
> If the mismatch were mainly caused by a global conservatism effect, then the **Global-Only** model should perform similarly to the **Full Per-Phrase** model. We conduct this experiment on GPT-4o:
>
> | Dataset | Global-Only | Full Per-Phrase | Gain |
> | --- | ---: | ---: | ---: |
> | gsm-hard | 0.5930 | 0.7418 | +0.1488 |
> | gsm8k | 0.5028 | 0.5945 | +0.0917 |
> | MedQA | 0.4996 | 0.5783 | +0.0787 |
> | average | 0.5522 | 0.6861 | +0.1339 |
>
> These results show that a single global conservatism parameter explains only a limited portion of the effect. Therefore, global conservatism effect is **not sufficient** to explain the reported human--LLM mismatch. The dominant signal remains marker-specific semantic divergence.
>
> **[W3/Q3: aggregation method]**
>
> As suggested, we conducted the following aggregation:
>
> - **max**: uses the most uncertain marker only;
> - **position-weighted**: upweights later markers;
> - **last-sentence**: uses markers from the final two sentences only;
> - **negation-aware**: flips markers under local negation;
> - **tf-weighted**: weights markers by frequency.
>
> (GPT-4o results)
> | Method | gsm-hard | gsm8k | MedQA | piqa | avg |
> | --- | ---: | ---: | ---: | ---: | ---: |
> | mean (baseline) | 0.7418 | 0.5945 | 0.5783 | 0.6117 | 0.6316 |
> | max | 0.7297 | **0.6237** | **0.5881** | 0.5776 | 0.6298 |
> | position_weighted | 0.7363 | 0.5455 | 0.5778 | 0.6252 | 0.6212 |
> | last_sentence | **0.7611** | 0.6205 | 0.5559 | **0.6462** | **0.6459** |
> | negation_aware | 0.6450 | 0.6172 | 0.5689 | 0.6040 | 0.6088 |
> | tf_weighted | 0.7345 | 0.5820 | 0.5822 | 0.6115 | 0.6276 |
>
> The main conclusion is that VOCAL is **not** sensitive to the averaging-based aggregation rule.
>
> **[W4/Q4: baseline comparison;gain isolation]**
>
> Verbal prompting is necessary for VOCAL because it relies on the model’s self-expressed uncertainty. At the same time, several baselines access signals VOCAL does not, such as token probabilities or multiple samples (e.g., SE), which are also stronger and more costly sources of information. We will revise the paper to clarify this trade-off.
>
> Figure 4 provides such a **prompt-matched** comparison to isolate the sources of gains, where (1) the two methods share the same prompt but (2) their lookup tables are different: human-sourced and VOCAL learned. We show that VOCAL consistently yields substantial AUROC improvements over $\mathcal{Q}_{VU-H}$ (e.g., boosting GSM8K with GPT-4o from 0.41 to approximately 0.60). The empirical evidence confirms that the prompt merely serves as an elicitation vehicle, whereas VOCAL’s learned mapping is the primary factor responsible for the performance gains.
>
> **[W5: related work positioning]**
> We will sharpen the Related Work by contrasting verbalized and probabilistic confidence, adding discussion on LLM self-knowledge and metacognition. We will position VOCAL in this context as an optimization-based bridge that aligns linguistic uncertainty markers with empirical correctness.
>
> **[Q2: uncertainty-marker extraction pipeline]**
>
> In our setup, the uncertainty signal comes from the model’s own self-expression. That means we prompt LLM to use verbal markers to describe the answer and output a verbal uncertainty marker list in a JSON file. There is no other extraction procedures applied.
>
> **[Q5: transferability across models]**
> To demonstrate the similarity of the learned lookup tables, we computed **shared-markers** rank correlations on the  learned lookup tables between each pair of models, via **Spearman’s ρ** and **Kendall’s τ**;
>
> | Pair | #Common | Spearman ρ | p-value | Kendall τ | p-value |
> | --- | ---: | ---: | ---: | ---: | ---: |
> | Qwen-72B vs DeepSeek | 99 | 0.5542 | 2.66e-09 | 0.4140 | 1.44e-09 |
> | GPT-4o vs DeepSeek | 127 | 0.4516 | 9.88e-08 | 0.3162 | 1.39e-07 |
> | GPT-4o vs Qwen-72B | 99 | 0.4291 | 9.37e-06 | 0.3007 | 1.07e-05 |
>
> These results suggest that models are **substantially aligned**: Spearman correlations range from **0.43 to 0.55**, and Kendall correlations range from **0.30 to 0.41**, all highly significant. However, these correlations are far from 1.0, so the rankings are **not identical**.
>
> Therefore, our take is that cross-model transfer is driven by **both**: a coarse monotonic pattern that is broadly shared across models (e.g., markers expressing stronger uncertainty tend to remain relatively high-uncertainty across models), and a nontrivial amount of ranking consistency among LLMs.

---

> > ### Author Rebuttal · Reviewer_fp15 · 2026-04-04
> >
> > Thank you for taking time to clarify concerns and answering questions. I will adjust the score.

---

> > > ### Author Response · Authors · 2026-04-04
> > >
> > > Thank you for your acknowledgement! We are glad to know that we have fully addressed your concern. We will include this discussion in our next revision.

---

### Official Review · Reviewer_7sEd · 2026-03-20

**Soundness:** 3
**Presentation:** 3
**Significance:** 3
**Originality:** 3
**Overall Recommendation:** 4
**Confidence:** 3

**Summary:**

The work studies whether verbal uncertainty markers such as “likely,” “possible,” and “almost certain” can serve as a usable uncertainty signal for LLMs. The authors first build a human uncertainty-marker lookup table from psychology and decision-science literature, aggregating 115 markers with numerical interpretations from an average of 336 human ratings per phrase. They then evaluate whether LLM verbal hedges align with those human meanings.  It shows that alignment is only partial: human-derived mappings give non-trivial uncertainty prediction, but often diverge from model behavior.

To address this mismatch, the paper proposes VOCAL, which learns model-specific numerical scores for uncertainty markers from model outputs and correctness labels. If a response contains markers with learned scores (C_i), the paper aggregates uncertainty and optimizes marker values with a binary cross-entropy objective plus graph-Laplacian smoothing over semantically similar markers. It demos that VOCAL can extract useful uncertainty from a single response, avoiding sampling at inference time.

Empirically, the paper reports that VOCAL improves over the human lookup table, beats single-sample baselines such as G-NLL/PPL in most reported settings. It is also often competitive with more expensive multi-sample UQ methods across several QA datasets and multiple LLMs. The paper also reports partial cross-LLM transferability of learned marker mappings.

**Compliance With Llm Reviewing Policy:**

Affirmed.

**Final Justification:**

The rebuttal is constructive. It strengthens the paper on implementation clarity and broadens the empirical scope with an initial long-form result. I appreciate the authors' effort very much! The rebuttal reinforced my prior assessment and I would continue to support the paper for publication.  Thanks!

**Key Questions For Authors:**

1. VOCAL currently operates on single-turn QA. Can the learned lookup transfer to chain-of-thought or multi-step agent trajectories without retraining, or does drift require per-turn recalibration? This directly addresses your own limitation on long-form generation.

2. How sensitive are the main comparisons to prompting protocol? Since VOCAL uses verbal-uncertainty prompting while other baselines use naive CoT, can you report matched-prompt comparisons or ablations showing that the gains persist under a common prompting setup?

3. What exactly is the deployment assumption for VOCAL’s training data? The method appears to optimize marker values using correctness labels. Do you assume access to a labeled calibration set for each model/domain, or is there a practical weak-supervision/self-evaluation variant?

4. You use string matching + the prompted “vue” field. How sensitive are VOCAL results to extraction errors or ambiguous phrases (e.g., “I think” appearing idiomatically)? A quick ablation on a noisy extractor would strengthen soundness claims.

**Limitations:**

The paper does discuss technical limitations such as coarse representation capacity, ambiguity in marker extraction, and domain/cultural variation in verbal uncertainty interpretation. However, the Impact Statement is kind of generic. It would be greatly appreciated, if the paper more explicitly addresses risks such as, mis calibrated verbal hedges increasing user over-trust, strategic or manipulative phrasing of uncertainty, cross-cultural mismatch in interpreting markers, and failure in high stakes domains like medicine if users treat verbal hedges as faithful probabilities, etc.

**Strengths And Weaknesses:**

[Soundness]

The optimization is provably convex (BCE + quadratic Laplacian), hyper-parameters are ablated, and experiments are carefully controlled (same prompts, greedy vs. temperature-0.8 sampling, 1 000 test questions per dataset). Correctness labels are objective (exact-match or multiple-choice). The authors report cases where VOCAL under-performs multi-sample methods (TriviaQA), show failure modes (under-/over-confidence), and note that marker extraction can be noisy. No cherry-picking; full cross-LLM transfer matrix and sample-efficiency curves are provided. Minor weakness: marker extraction is rule-based/string-matched rather than a trained extractor, but this is acknowledged.

[Presentation]

I think the narrative is clear: gap → human baseline → divergence discovery → VOCAL → results → limitations. Figure 1 instantly communicates the core idea. Table 2 (human vs. GPT-4o probabilities) is the most striking single result in the paper. Equations are cleanly typeset, prompts and the full 115-marker table are in appendices, and every baseline is precisely cited with implementation details. Minor suggestion: the cross-LLM transfer figure could include standard deviations; otherwise, an expert can reproduce everything from the provided appendices.

[Significance]

Verbal UQ is the only interface users actually see. Current sampling-based methods (SE, Deg) add latency and cost that block real-time deployment in medicine, education, or chat. VOCAL delivers near-SOTA discrimination from a single response, directly improving trustworthiness at negligible cost. The public UM-Lookup table is a reusable foundation for future metacognition, alignment, and human-AI interaction work. Even modest AUROC gains (e.g., +0.2 on GSM8K) on reasoning-heavy tasks matter because they reduce hallucinations precisely where users need reliability most. The significance is somewhat tempered by scope. The evaluations are mostly on short-form QA benchmarks, whereas the motivating discussion is about broader generative trustworthiness and hallucination. Extensions to long-form generation are left as future work.

[Originality]

I think the paper is original in framing, even if the optimization itself is simple. The strongest novelty is not the BCE-plus-regularization objective by itself, but the combination of:
1. a human verbal-uncertainty resource,
2. an explicit human-vs-LLM semantic mismatch analysis, and
3. a lightweight model-specific remapping procedure.
While this is not a wholly new class of UQ algorithm, it does offer a new perspective on linguistic uncertainty as a semantics-learning problem rather than only a prompting or calibration problem. That is enough for me to regard the paper as clearly original, though not radically so.

---

> ### Author Rebuttal · Authors · 2026-03-31
>
> We thank you for your insightful review!
>
> **[W1/Q4: marker extraction pipeline]**
>
> In our setup, the main signal comes from the model’s **explicit self-reported uncertainty** in the structured vue field, rather than unrestricted phrase mining from the full answer. This reduces ambiguity from idiomatic cases such as “I think,” since the model is asked to list the uncertainty markers it intentionally used. That said, extraction noise is still possible: the model may omit markers, list them imprecisely, or use phrases with ambiguous scope. We will clarify this limitation and add a small robustness check by comparing structured vue field with a noisier extraction setting that directly matches uncertainty phrases in the raw answer text.
>
> **[W3/Q1: long-form dataset results]**
>
> As suggested, we extended our evaluation to long-form dataset LongFact. We follow the same setting as in our manuscript and conduct experiment on DeepSeek-V3.1. AUROC Results are summarized as
>
> | Dataset | Model | G-NLL | PPL | SE | VOCAL |
> | --- | --- | ---: | ---: | ---: | ---: |
> | LongFact| DeepSeek-V3.1 | 0.4012 | 0.5912 | 0.4037 | 0.6719 |
>
> Long-form UQ remains highly challenging. Traditional methods like G-NLL and SE struggle significantly (AUROC < 0.5). Because their cumulative log-probabilities suffer from severe length bias in extended sequences, with PPL performing only slightly better.
>
> **[Q2: Prompt-matched comparisons]**
>
> Figure 4 provides such a **prompt-matched** comparison to isolate the sources of gains, where (1) the two methods share the same prompt but (2) their lookup tables are different: human-sourced and VOCAL learned. We show that VOCAL consistently yields substantial AUROC improvements over $\mathcal{Q}_{VU-H}$ (e.g., boosting GSM8K with GPT-4o from 0.41 to approximately 0.60). The empirical evidence confirms that the prompt merely serves as an elicitation vehicle, whereas VOCAL’s learned mapping is the primary factor responsible for the performance gains.
>
> **[Q3: Calibration data requirements]**
>
> VOCAL assumes a labeled calibration set for each target setting. We use this held-out data to learn a model-/setting-specific lookup table, and then directly apply that table at test time without additional sampling or retraining. In the current paper, we do not assume a fully label-free variant; a weak-supervision or self-evaluation extension is an interesting future direction, but it is beyond the scope of the present work.

---

> > ### Author Rebuttal · Reviewer_7sEd · 2026-04-06
> >
> > // First of all I am really sorry for the delay, while traveling b2b across time zones and losing my track (often times w/o vpn access to the review system - sigh..)
> >
> > The rebuttal is constructive and I appreciate it. It meaningfully strengthens the paper on implementation clarity and broadens the empirical scope with an initial long-form result. It does not 100% resolve the methodological comparison issue, and it leaves the confidence/uncertainty notation and objective formulation partially unanswered.  However, as nothing is perfect, I would continue to support the paper yet not revise my assessment upward dramatically.

---

### Decision · Program_Chairs · 2026-04-30

**Decision:**

Accept (regular)

**Comment:**

**Summary**
This paper studies how LLMs diverge from humans in quantifying verbal uncertainty. This paper curates a corpus of human uncertainty markers from psychology and decision-science literature, and benchmark LLMs against it. This benchmark shows that LLMs encode verbal uncertainty with numerical levels that substantially differ from those of humans. Then, this paper introduce VOCAL, a novel optimization-based algorithm that learns an optimal uncertainty profile over uncertainty markers directly from LLM outputs. VOCAL enables a direct, marker-level comparison of confidence semantics between humans and LLMs, disentangling mismatch and revealing systematic confidence disparities in verbal expressions.

**Strengths**
- The problem framing is novel, insightful, and previously under-explored.
- Unlike traditional UQ methods, VOCAL is more lightweight.
- Empirical results show competitive results across QA benchmarks.
- The UM-lookup table is a valuable contribution for future research.

**Weaknesses**
- VOCAL is a supervised method, but comparisons are mainly done against unsupervised UQ methods. The author response added comparisons.
- Prompting format between VOCAL and baselines appear different. In response, the authors provided prompt-matched comparisons and showed consistent improvements.
- The tasks are mostly short-form binary or multiple-choice QA. In responses, the authors added results for a long-form dataset.
- The main metric being shown is AUROC. In the response, results using the ECE metric is also reported.

Overall, this paper considers an under-explored problem, presents insightful analysis and competitive results, and contributes valuable artifacts. The weaknesses are addressed in responses. I think this paper is a clear accept.